# Synthesis, Sorting, and Applications of Single-Chirality Single-Walled Carbon Nanotubes

**DOI:** 10.3390/ma15175898

**Published:** 2022-08-26

**Authors:** Marianna V. Kharlamova, Maria G. Burdanova, Maksim I. Paukov, Christian Kramberger

**Affiliations:** 1Centre for Advanced Material Application (CEMEA), Slovak Academy of Sciences, Dubrávská cesta 5807/9, 854 11 Bratislava, Slovakia; 2Institute of Materials Chemistry, Vienna University of Technology, Getreidemarkt 9-BC-2, 1060 Vienna, Austria; 3Laboratory of Nanobiotechnologies, Moscow Institute of Physics and Technology, Institutskii Pereulok 9, 141700 Dolgoprudny, Russia; 4Center for Photonics and 2D Materials, Moscow Institute of Physics and Technology, 9, Institutsky Lane, 141700 Dolgoprudny, Russia; 5Institute of Solid State Physics, Russian Academy of Sciences, 142432 Chernogolovka, Russia; 6Faculty of Physics, University of Vienna, Strudlhofgasse 4, 1090 Vienna, Austria

**Keywords:** carbon nanotubes, sorting, synthesis, separation, chirality

## Abstract

The synthesis of high-quality chirality-pure single-walled carbon nanotubes (SWCNTs) is vital for their applications. It is of high importance to modernize the synthesis processes to decrease the synthesis temperature and improve the quality and yield of SWCNTs. This review is dedicated to the chirality-selective synthesis, sorting of SWCNTs, and applications of chirality-pure SWCNTs. The review begins with a description of growth mechanisms of carbon nanotubes. Then, we discuss the synthesis methods of semiconducting and metallic conductivity-type and single-chirality SWCNTs, such as the epitaxial growth method of SWCNT (“cloning”) using nanocarbon seeds, the growth method using nanocarbon segments obtained by organic synthesis, and the catalyst-mediated chemical vapor deposition synthesis. Then, we discuss the separation methods of SWCNTs by conductivity type, such as electrophoresis (dielectrophoresis), density gradient ultracentrifugation (DGC), low-speed DGC, ultrahigh DGC, chromatography, two-phase separation, selective solubilization, and selective reaction methods and techniques for single-chirality separation of SWCNTs, including density gradient centrifugation, two-phase separation, and chromatography methods. Finally, the applications of separated SWCNTs, such as field-effect transistors (FETs), sensors, light emitters and photodetectors, transparent electrodes, photovoltaics (solar cells), batteries, bioimaging, and other applications, are presented.

## 1. Introduction

The universal architecture of all SWCNTs is a rolled-up stripe of hexagonal sp^2^ carbon forming the seamless wall of an extended cylinder. The ends of SWCNTs are typically closed by curved caps, resembling half spheres of fullerenes. The required Gaussian curvature can be achieved by inserting six pentagons at either end of the SWCNTs. Due to the enormous aspect ratio (>100), the properties of SWCNTs are solely dependent on the walls of the cylinder and not the terminating caps.

The detailed atomic structure of the sidewalls uniquely determines all properties of a SWCNT. The full atomic structure is in turn fully defined by the so-called chiral vector (C_h_) running along the circumference. While C_h_ connects two different equivalent carbon atoms in the graphene lattice, in the SWCNT, it describes circumferential periodicity connecting each atom with itself [1]. The graphene lattice vector is given by two integer indices (n,m) counting unit cells along the crystallographic (1,0) and (0,1) directions. These directions run through the centers of adjacent hexagons. The discrete possible lengths of C_h_ define the discrete set of possible diameters for SWCNTs. The twisting of the hexagons on the surface of the SWCNTs is determined by the so-called chiral angle θ, which is measured between C_h_ and the (1,0) direction [1].

SWCNTs with C_h_ of the form (n,0) or for short (n,0) SWCNTs have a chiral angle of θ = 0°. At every atom, one of the bonds is parallel to the axis, and the other two bonds form a closed zigzag loop around the circumference. They are called zigzag SWCNTs. For (n,n) SWCNT θ = 30° and at every carbon atom, there is a bond aligned along the circumference. The bonds closest to a given circumference or position on the SWCNT axis form an armchair path. (n,n) SWCNTs are called armchair SWCNTs. All other SWCNTs with 0 < θ < 30 are called chiral SWCNTs [1]. The chiral vector (n,m) also fully determines the electronic properties of SWCNTs by determining the exact boundary conditions for the zone-folding scheme on the band structure of graphene. If n-m is a multiple of three, the Dirac point of graphene is included in the band structure of the SWCNT, and it becomes a metal. Otherwise, the Dirac point is excluded, and the SWCNT is a semiconductor where the bandgap scales with the inverse diameter. All armchair and one-third of zigzag and chiral SWCNTs are metals, while the rest are all semiconductors with different bandgaps [1].

There are three basic processes used to create SWCNTs: arc discharge, laser ablation, and chemical vapor deposition (CVD). The bundled nanotube samples with a large quantity of catalyst residuals can be obtained as powdered samples using the arc discharge and laser ablation techniques [1,2,3]. In contrast, the CVD method allows preparing the samples of powdered or well-aligned bundled or individual nanotubes with or without substrates that contain a small amount of impurities [1,2,3]. Several CVD approaches were developed for the synthesis of high-quality SWCNTs: thermal CVD with fixed [4,5,6,7,8,9,10,11,12,13] or fluidized [9,12,13,14,15,16,17,18] bed reactors, plasma-enhanced CVD, aerosol CVD [19,20,21,22,23,24,25,26], and floating catalyst CVD [27,28,29,30,31,32,33,34]. It was shown that varying synthesis parameters changes the properties of obtained nanotubes (diameter), their quality, and amount. In a typical synthesis process, the mixture of metallic and semiconducting SWCNTs is present.

This review is dedicated to the chirality-sensitive synthesis and separation of SWCNTs toward applications on a large scale, as presented in Figure 1. The first part of the review is dedicated to the discussion of synthesis methods of SWCNTs with semiconducting and metallic conductivity-type and single-chirality SWCNTs. Section 2.1 discusses the epitaxial growth method of SWCNT (“cloning”) using nanocarbon seeds. Section 2.2 describes the growth method using nanocarbon segments obtained by organic synthesis. Section 2.3 presents the catalyst-medicated chemical vapor deposition synthesis. The second part of the review presents a discussion of the separation of SWCNTs. Section 3.1 includes the separation of SWCNTs by conductivity type, such as electrophoresis (dielectrophoresis), density gradient ultracentrifugation (DGC), low-speed DGC, ultrahigh DGC, chromatography, two-phase separation, selective solubilization, and selective reaction methods. Section 3.2 describes the single-chirality separation of SWCNTs, such as the density gradient centrifugation (DGC), two-phase separation and chromatography methods. Section 4 describes the applications of separated SWCNTs, such as field-effect transistors (FETs), sensors, light emitters and photodetectors, transparent electrodes, photovoltaics (solar cells), bioimaging, and other applications.

## 2. Synthesis Methods of Semiconducting, Metallic, and Single-Chirality SWCNTs

### 2.1. Catalyst-Controlled CVD Synthesis of Single-Chirality SWCNTs

Controlling the size and shape of catalytic particles before they eventually nucleate and grow an extended SWCNT during CVD synthesis is the most direct route to predetermine the diameter or even chirality.

There are numerous articles reported on the chirality-sensitive synthesis of SWCNTs on a wide variety of catalyst materials. They comprise CoMo [35,36,37,38,39], FeCo [40], NiFe [41], Co [42,43,44], FeCu [45], Au [46], CoMn [47], Ni [48], Fe [49,50], CoPt [51], Co_x_Mg_1−x_O [52], CoSO_4_, WCo alloy [53,54,55], and Mo_2_C [56]. These diameter-controlling catalysts are used to grow SWCNT from different carbon feedstocks: CO [36,37,38,39,42,43,44,45,47,48,49,51,52], CH_3_OH [38], CH_4_ [42,50] and C_2_H_2_ [41].

The pioneering report on catalyst-controlled chirality selectivity in the CVD synthesis of SWCNTs is from 2003. In [46], SWCNTs were synthesized from a CoMo catalyst supported on SiO_2_ from CO as carbon source at 750 °C. The obtained SWCNT sample contained two dominant chiralities—(6,5) and (7,5). These two make up 57% of the semiconducting fraction or 38% of the entire samples. A colored contour plot of excitation-normalized fluorescence intensities and an (n,m)-resolved intensity map for the sample are shown in Figure 2. The linewidth of each hexagonal tile in the Hamada map is proportional to the observed intensity for this (n,m) SWCNT (Figure 2b).

Reports on the effects of synthesis parameters, such as the gaseous carbon precursor, its pressure [37], catalyst composition [41], type of support [37,39], and synthesis temperature [39,40,41,43,45,47,51,52] followed soon.

Either CO or CH_4_ was used as gaseous carbon source to grow chirality-enriched SWCNTs from a binary CoMo catalyst [39]. When all other parameters were kept the same and only CO or CH_4_ was used, the two gases favored very different chiralities. Figure 3a,b shows optical absorption and the chirality map of CO-SWCNT and CH_4_-SWCNT. Evidently, CH_4_ is far less selective, and the CO-synthesized SWCNTs are dominated by the (6,5) chirality. The chirality distribution obtained with CH_4_ extends to larger diameters and appears to be clustered towards either an armchair or zigzag direction. This stark difference was attributed to the role of hydrogen gas, which is only formed in the decomposition of CH_4_ but not in that of CO. H_2_ boosts the reduction rate of the CoMo particles and also slows down the nucleation rate at which the seeding carbon island detaches from the catalyst particle. Every time the nucleation is delayed, more initial carbon is available and can then eventually nucleate an energetically favored larger diameter cap.

A CoMo catalyst was also used to compare the effects of four different carbons [38]. The four feedstocks were: CO, C_2_H_5_OH, CH_3_OH, and C_2_H_2_. The pressure of the carbon precursors was identified as an important factor to control over chiralities. Tightly clustered (n,m) SWCNTs were synthesized under CO and vacuumed C_2_H_5_OH and CH_3_OH. The clustered chiralities were always close to the armchair direction.

Figure 4 presents photoluminescence maps belonging to SWCNTs obtained from the four different carbon sources. CO yields the narrowest (n,m) distribution centered at the (7,6), (7,5), and (8,4) tubes. C_2_H_5_OH and CH_3_OH lead to (8,6), (9,5), and (8,7) tubes. The findings in [38] unequivocally demonstrate that the carbon precursor chemistry is an important part of the growth mechanisms of tightly clustered (n,m) SWCNTs.

In [57], the SWCNTs were synthesized using a CoMo catalyst and CO carbon feedstock. The authors observed a clear dependence between the investigated CO pressure and the chirality selectivity. In the pressure range varying from 2 to 18 bars, the predominant relative yields of (6,5), (7,5), and (7,6) SWCNTs were observed. It was proved using photoluminescence (PL) intensity maps obtained for SWCNT grown under different CO pressures. In Figure 5a, the peaks corresponding to (6,5), (7,5), and (7,6) are marked [37]. In particular, 18 bar results in the (6,5) dominant formation, and its intensity decreases monotonically with the pressure. In contrast, the intensity of the (7,6) nanotube exhibited the opposite behavior. It is dominant at a pressure of 2 bar, and its intensity decreases with increasing pressure. The (7,5) PL peak intensity was highest at 12 bar CO (see Figure 5a). To evaluate this change, the relative abundance of the chiralities at different CO pressures was presented (Figure 5b). Going from 2 to 18 bar, the abundance of (6,5) increased from 2% to 48%, while the (7,6) abundance dropped from 26% to 7%. The (7,5) abundance peaked at 28% and 12 bar CO. In support, the optical absorption spectroscopy measurement was taken. Figure 5c shows spectra of the SWCNTs produced under different CO pressures. The first and second absorption peaks of a semiconducting SWCNT undergo changes in their relative intensities according to the evolving PL and relative abundances [37].

In another research, the observed chirality distribution of synthesized SWCNTs was tuned by Ni_x_Fe_1-x_ catalytic nanoparticles [41]. In detail, the synthesis process was conducted using C_2_H_2_ as carbon source at 600 °C. Figure 6 shows the PL intensity of different chirality distributions obtained for the synthesized samples. These data show Ni nanoparticle chirality distribution with predominant (9,4) tubes and small amounts of (8,4), (7,5), (10,2), (8,6), (9,5), and (10,3) tubes. When a Ni_0.67_Fe_0.33_ catalyst was used, similar results were achieved, and the distribution with an enriched amount of (7,6) tubes was obtained. Samples synthesized with Ni_0.5_Fe_0.5_ and Ni_0.27_Fe_0.73_ catalysts are characterized by significant changes in SWCNT distributions. Ni_0.5_Fe_0.5_-catalyzed SWCNT samples contain mainly (7,5) and (8,4) tubes and a small fraction of (7,6), (8,3), and (6,5) tubes. A Ni_0.27_Fe_0.73_ catalyst results in a much narrower distribution of predominantly enriched (8,4) and smaller fractions of (7,5), (6,5), (7,6), and (8,3) tubes. Authors suggested that changes of catalyst nanoparticle stoichiometry result of compositional tuning affect the lattice match of the catalyst with certain tube chiralities.

Different morphology catalysts, SiO_2_ and MgO, showed a significant difference on the distribution of SWCNT chirality [39]. This resulted in the growth of different SWCNTs with a near-armchair structure. SiO_2_ catalysts result in the predominant (6,5) formation of SWCNT. In contrast, samples synthesized using MgO showed fewer (6,5) tubes and more (7,5), (8,4), and (6,6) tubes.

The temperature dependence on the synthesis was also investigated using different catalysts. Overall, the increase in the temperature led to an increase in the SWCNT diameters and a broadening of the chirality distribution [39,40,41,43,45,47,51,52]. Almost single chirality (6,5) SWCNT samples were observed at temperatures of about 500–700 °C. In [58], SWCNT was synthesized using an FeRu bimetallic catalyst supported on silica and CH_4_ as carbon source at different temperatures. Figure 7a–c shows the change in the photoluminescence maps for SWCNTs synthesized at 600, 700, and 850 °C. The SWCNTs synthesized at 600 °C consist of a (6,5) tube with a relatively high intensity of PL and weak peaks from other chiralities. In contrast, the increase in abundance of (7,6), (7,5), and (8,4) tubes was observed at 700 °C. In the sample obtained at 850 °C, the (8,4) tube dominated. In the last case, significant amounts of the (7,5), (7,6), (8,6), and (9,4) tubes were present (Figure 7) [58].

A narrower chirality distribution was reported in [51]. The bimetallic CoPt catalyst was discussed for the selective growth of the (6,5) tubes at temperatures of about 800–850 °C. As a possible explanation, the formation of CoPt alloy and its high stability were discussed to be responsible for the growth of smaller-diameter SWCNTs with a narrow distribution.

Recently, the selective growth of single-chirality SWCNT different from (6,5) was reported in [7,43,49,53,54,56,59]. In [43,59], near-armchair SWCNTs with a chirality of (9,8) were shown. The authors in [43] synthesized the single-chirality SWCNT samples. This resulted in 59.1% of (9,8) chirality in semiconducting SWCNTs. It was explained by a strong metal–support interaction, which stabilized the Co clusters with a narrow diameter distribution, which resulted in a selective growth of the (9,8) tubes. A similar catalyst, CoSO_4_ supported by SiO_2_, showed a selective growth of (9,8) nanotubes [59]. A 51.7% abundance among semiconducting SWCNTs was observed. This effect was explained by the formation of Co particles, which precisely matched the (9,8) tube diameter. Sulfur present in the synthesis was suggested as a precursor, which limited the agglomeration of Co particles, resulting in Co-S composition. This consequently enabled the selective growth of (9,8) tubes. Figure 8a shows photoluminescence intensity maps of SWCNTs dispersed in 2 wt% sodium dodecyl benzene sulfonate (SDBS) D_2_O solution. Figure 8b shows that the catalyst is responsible to the (9,8) tube (51.7%) after 540 °C reduction. In addition, a low-relative-abundance (>3%) SWCNT was detected. Among them, (9,7), (10,6), (10,8), (8,7), (10,9), and (6,5) were observed. The occurrence of such species shows a strong selectivity toward high chiral angle tubes in SWCNT growth, which is consistent with earlier research. The histogram in Figure 8b presents SWCNTs’ abundance using PL (blue), Raman (red), and absorption (yellow) [59]. This SWCNT CoSO_4_/SiO_2_ was obtained after catalyst reduction at 540 °C.

The aerosol floating-catalyst CVD technique was employed to create large-diameter SWCNTs with a narrow (n,m) distribution and dominant (13,12) tubes (d = 1.67 nm) utilizing ferrocene as the catalyst precursor [49]. Over 90% of the obtained SWCNTs were near armchair. The authors explained it through a strong etchant of ammonia. Due to the greater reactivity and decreased stability when compared with high chiral angle tubes, it selectively etched off SWCNTs with small chiral angles. Due to their increased curvature, tiny-diameter nanotubes had a similar effect and were removed. The presence of NH_3_ could also affect the catalyst clusters already during nucleation. This suppresses the growth of tubes with small chiral angles. In a 500 ppm NH_3_ sample, there were 108 individual SWCNTs, and in a 0 ppm NH3 sample, there were 95 nanotubes, according to electron diffraction analysis. This result is depicted in (n,m) maps for samples grown in both conditions (Figure 9) [49]. The 500 ppm NH_3_ sample contained in total 37 different chiral structures with three main chiralities, (13,12), (12,11), and (13,11), and abundances of 13, 8, and 8, respectively. This constitutes about 30% of the nanotubes. The majority of chiralities were concentrated near the main chirality of semiconducting (13,12) nanotubes. In a 0 ppm NH_3_ sample, a broader distribution of chiralities was observed. Over 95 of the investigated nanotubes had 52 different chiralities with (12,10) showing slightly higher abundance [49].

Recently, the chirality-selective growth was achieved by the optimization of the SWCNT–catalyst interaction. The authors in [53,54] used WCo alloy catalysts and selected the specific structure as a template to realize the chirality-controlled growth of SWCNTs. Using ethanol as carbon source, the (12,6) tubes (d = 1.28 nm) with an abundance of more than 92% were specifically created [53]. Such high abundance was achieved using W_6_Co_7_ alloy nanoparticles as catalysts. The observed chirality enrichment was most likely the result of a strong structural match between the placement of the metal atoms in the nanocrystal catalyst and the placement of the carbon atoms around the circle of the nanotube. The selective chirality growth was confirmed by micro-Raman measurements. Clear radial breathing mode (RBM) peaks centered around ~197 cm^−1^ were observed. The corresponding G-mode bands at ~1580 cm^−1^ exhibited typical Breit–Wigner–Fano line shapes, a feature typical of metallic SWCNTs. After a comparison with s Kataura plot, it was assigned to the chirality of (12,6). The electron diffraction measurements on solitary tubes grown across silicon substrate slits and with RBMs at 197 cm^−1^, which all display typical electron diffraction patterns indexed to (12,6), were used to further confirm this assignment. The content of (12,6) tubes was 92.5%, according to the ultraviolet–visible–near-infrared (UV–VIS–NIR) absorption spectra of aqueous dispersion obtained from 42 SWCNT samples. Based on approximately 3300 detected tubes, the statistics of the RBMs in the micro Raman spectra and counting for the coverage of excitation yielded an estimate of 94.4%. Consequently, it was shown utilizing three distinct ways that it was possible to achieve the direct growth of SWCNTs with a single dominant chirality in a proportion greater than 92% [53].

The similar catalyst W_6_Co_7_ was used for the growth of zigzag SWCNT. The authors in [54] synthesized (16,0) SWCNT with an abundance of ~80%. Due to the structural similarity between the open end of the tube and [50] the groupings of metal atoms in the (1 1 6) planes of the catalyst, it was hypothesized that the nanocrystal catalyst’s (1 1 6) planes served as templates for the (16,0) tubes [50]. The structural match between the tubes and the nanocrystal catalyst was assigned to the thermodynamic ascendancy for the growth of SWCNTs with specific chiralities as well as the growth kinetic. They therefore believed that the combination of the catalyst’s structural template effect and growth kinetics optimization can be used to manufacture zigzag SWCNTs in a dominant manner. Similarly, in [50], the CVD synthesis of near-zigzag SWCNTs with a dominant chirality of (15,2) using an Fe catalyst succeeded. It was created utilizing ethanol bubbler operation at 1050 °C, 150 cm^3^ min^−1^ H_2_, and 200 cm^3^ min^−1^ Ar as carbon feeding stock. The peak intensity for the SWCNTs in radial breathing mode (RBM) was at 195 cm^−1^ (under excitations of both nearly resonant 532 and 785 nm laser excitation). Indicative of semiconducting zigzag, the associated tangential vibration (G mode) bands at 1583 cm^−1^ showed single Lorentzian symmetric line forms. They also observed that Raman active modes corresponding to oTO phonon branches all appeared at the same position. In addition, chirality assignment was performed by photoluminescence spectroscopy. The clear fluorescence signal from samples grown on the silicon sphere with RBMs at ~195 cm^−1^ indicated a semiconducting type of conductivity of such SWCNTs. The initial van Hove optical transition energy (E_11_) of the (16,0) SWCNT was represented by the fluorescence emission at around 0.77 eV under both 532 and 785 nm excitations. The beam diffraction of the SWCNTs was used for the final conformation of the chirality assignment. As a result, a bundle made up of two SWCNTs that were grown across the silicon substrate’s slits at an RBM frequency of about 195 cm^−1^ exhibited a characteristic electron diffraction pattern that was indexed to a (16,0) SWCNT. The micro-Raman spectra were used to estimate the abundance of (16,0) SWCNTs in the samples developed at 1050 °C under optimal CVD conditions at 79.2%. Overall, these findings amply illustrated the unique W6Co7 catalyst’s direct development of zigzag (16,0) SWNTs [50].

### 2.2. The Epitaxial Growth Method of SWCNT Using Nanocarbon Seeds

Most of the sorting techniques require the usage of chemicals, which have to be removed for certain applications. Moreover, a successful sorting can be achieved by using individualized SWCNT after pretreatment, which involves ultrasonication. This results in the introduction of defects and impurities. However, the majority of them are not selective for diameter and chirality separation and require multistep separation for improved purity. One way to avoid these problems is to ‘seed’ the SWCNTs from another SWCNT or carbon structures. With the presorted SWCNTs by using chromatography, centrifugation, and so on, fullerenes can be then used to initiate the growing process, which can be called ‘cloning’. The cloning growth routes using pure carbon systems suffer from a catalyst particle, which become critical for many applications. Mimicking the epitaxial growth of a single SWCNT from nanocarbon fragments is one of the most promising techniques for batch synthesis of SWCNTs without a perfect control of the catalyst morphology and size, which is the most challenging synthesis part.

The short segments of the existing SWCNT with the known diameters and chirality can be used as a starting material (Figure 10a). To initiate the cloning process, the catalyst nanoparticles, such as Fe, are attached to each end of the opened tube. Then the carbon source is introduced at a certain temperature and pressure, and the growing of a new SWCNT with predominant diameters is observed. These new segments of SWCNTs at both sides can reach a few micrometers in length. In particular, the atomic force microscopy (AFM) images of the obtained structure confirmed the same predetermined diameters as the initial nanotube seeds (Figure 10b). However, the AFM technique did not give any information about the chirality, and the small difference between different chirality nanotubes with a similar diameter cannot be resolved. Later on, additional optical techniques, such as Raman spectroscopy, were implemented to show the similarity not only in the diameter but also in chirality. By using e-beam lithography, SWCNTs can be divided into manageable pieces. The open-end SWCNTs served as the starting material for the synthesis of SWCNTs [60]. However, in the earliest articles, the yield of the process was relatively low and did not exceed 40%. This is particularly due to the docked part, which is smaller in the diameter than the seeded part, also called parent part.

Next, Liu et al. used chirally enriched semiconducting and metallic (7,6), (6,5), and (7,7) SWCNTs produced by DNA-based gel chromatography [61]. They developed a catalyst-free cloning growth protocol that enables the production of long horizontally aligned SWCNTs. Importantly, they showed that annealing is a critical step for SWCNT cloning. This step was attempted to remove the defective edges from SWCNT sides, which terminated the growing processes. In particular, the authors showed that this method can be implemented to produce a single-SWCNT back-gated transistor and can be further used for circuit applications. A detailed understanding of the cloning mechanisms was achieved in [62]. For the vapor-phase epitaxial growth of seven single-chirality nanotubes of (9,1), (6,5), (8,3), (7,6), (10,2), and (6,6), they reported chirality-dependent growth kinetics and termination mechanism (7,7). Their findings showed that the active lifetimes of growth exhibit the opposite pattern from the growth rates of nanotubes, which rise with their chiral angles. They also observed that WCNTs, such as (6,6), had substantially shorter saturated lengths (about 1/3) than (6,5).

**Figure 10 materials-15-05898-f010:**
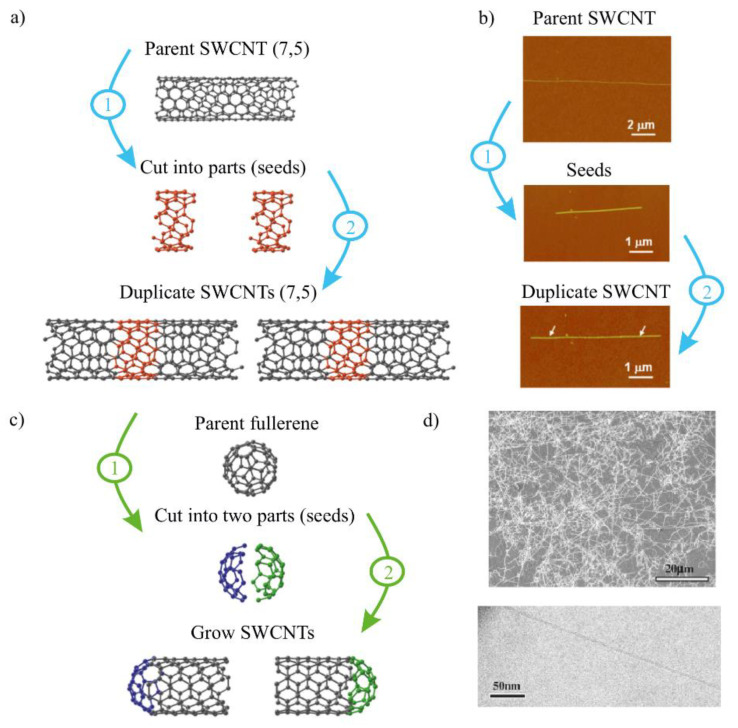
(**a**) Schematic of a cloning for the growth of identical SWCNTs from SWCNTs. At the first step, as-grown ultralong single-chirality SWNTs cut into shorter segments, seeds (red in color). At the second step, the grown SWNTs could be grown from the obtained seeds with activated edges. (**b**) The experimental implementation of the cloning method presented on figure (**a**). The duplicate SWNTs continued to grow from the seeds obtained from a parent SWCNT [60]. Copyright 2010, American Chemical Society. (**c**) Two-step cloning method of identical SWCNTs from fullerenes. The parent fullerene is cut in half. Each of the obtained parts is used to grow new SWCNTs. (**d**) Representative SEM image (**top**) and TEM image (**bottom**) of the SWCNTs grown from fullerenes on a SiO_2_/Si substrate [63]. Copyright 2003, with permission from Elsevier.

Takagi et al. showed that diamond nanoparticles can act as seeds for SWCNT growth [64]. This opens another route for SWCNT growth mechanism—the opening of fullerenes and using them as nucleation caps for SWCNTs [63]. The carbon cage of fullerenes can be selectively opened at the broken C–C bond under specific thermal oxidation and annealing conditions, which has limited options for carbon caps. Therefore, the diameter distribution of caps derived from fullerenes is not continuous but is instead discrete [65]. This makes fullerenes highly attractive for SWCNT growth. Additionally, it was demonstrated that developing small-diameter SWNTs is favored by higher fullerene opening temperatures, whereas growing large-diameter SWNTs is favored by lower temperatures. In comparison with the catalyst-grown SWCNTs, the fullerene-based SWCNTs have a steplike distribution of the diameters, which then can be fine-tuned [66].

Fullerenes can serve as the seed material for SWCNTs, as demonstrated by Ibrahim et al. Additionally, the authors of this article demonstrated how crucial the initial preparation and treatment stages are for the catalyst-free growth of NTs. Oxidation stages are demonstrated to be particularly important, and oxygen-based functional groups seem to be significant. When employing precursors devoid of oxygen, there is no growth in the final stage. Surprisingly, SWCNTs make up the majority of the synthetic tubes. However, it is also possible to obtain DWCNTs [67]. Sanchez-Valencia and coworkers reported the synthesis of the polycyclic hydrocarbon precursor C_96_H_54_, which was used for a short-capped (6,6) tube as seeds that were elongated into (6,6) SWNTs.

The poor activity of the nanocarbon seeds, which affects the growth rate and production of SWNTs, is one of the issues with SWCNT growth. One effective microwave-assisted route for producing SWNTs from carbon shards on a SiO_2_/Si substrate was demonstrated in [68]. Rapid heating from microwave irradiation helped to break apart polar groups attached to the SWCNTs and lessen the spontaneous closing of the open ends of the tubes. After plasma treatment, the SWNT that had survived and the carbon fragments attached to it were simply microwaved and used as a template for regeneration.

Despite the advancements in SWCNT cloning, the majority of forecasts indicate that only SWCNTs with large chiral angles will have higher possibilities for effectively copying their structures. The majority of the aforementioned techniques show that it is possible to reproduce SWCNTs from their own copies, but because the initial seeds might vary in chirality, it is still challenging to produce single-chirality SWCNTs. Moreover, there is a lack of statistical distribution of the cloning SWCNTs published so far, making the conclusion about the success of the SWCNT cloning difficult to estimate. All these problems limit the l-scale production of chirality-controlled SWNTs using seeds.

### 2.3. The Growth Method Using Nanocarbon Segments Obtained by Organic Synthesis

The progress achieved in “cloning” SWCNTs by using nanocarbon structures results in the development of a new direction—synthesis of SWCNT templates through organic routes. Some organic carbon nanostructure templates, such as nanorings, nanohoops, nanobelts, and nanoribbons, have been successfully used for SWCNT synthesis [69] (Figure 11). The dimensions and structure of these templates dictate the diameter and chirality of the resulting SWCNTs.

In [9,12,18,70], cycloparaphenylene nanorings were first synthesized and characterized. These structures can be used to grow the extended specific chirality SWCNTs without the need for high temperature. Due of their armchair SWCNT theme, aromatic belts have become more and more popular to synthesize. By joining the 2- and 11-positions with an eight-atom aliphatic bridge, a teropyrene system that structurally resembles a (8,8) armchair SWCNT was created [71]. A short (5,5) SWCNT was grown from corannulene [72]. In particular, Steinberg and Scott developed Diels–Alder cycloaddition for both hemispherical end caps and aromatic belts [73]. While hemispherical end caps cause SWCNTs to be closed at one end and open at the other, aromatic belts cause SWCNTs to have both ends open.

The SWCNTs with a defined diameter could be obtained by using cyclo(para-phenelene)s [74]. Later on, following this article, acene-inserted cyclo(para-phenelene)s were suggested as a source of segments of all armchair, chiral, and zigzag SWCNTs [75]. The important conclusions were made after cyclo(para-phenelene) synthesis of SWCNTs. CNTs are not formed without CPPs, and the diameters of grown CNTs match those of the CPPs employed [76].

The cyclodehydrogenation reaction on the Pt surface approach was also established to obtain structurally controlled SWCNT caps. Two types of SWCNTs were synthesized—armchair SWCNTs with (6,6) and zigzag SWCNTs with (9,0) nanotubes [69]. The ability to produce these kinds of SWCNTs with excellent purity using high-temperature sublimation was demonstrated. Results from Raman spectroscopy and scanning tunneling microscopy (STM) demonstrated that each SWCNT has a (6,6) chirality. Additionally, Zhou et al. employed C50H10, a predesigned polycyclic hydrocarbon precursor, as the cap of the (5,5) SWCNT. With a mean diameter of 0.82, the pure semiconducting SWCNTs were successfully produced. It is suggested that one of the main causes of the almost exclusive growth of semiconducting SWCNTs is the short diameter of C50H10 molecules, which causes a diameter-dependent growth of SWCNTs [77]. Several polycyclic hydrocarbon precursors were successfully used for the synthesis of six precursor molecules for large-diameter hemispherical buckybowls for the controlled synthesis of zigzag SWCNTs [78]. Based on these templates, they demonstrated the formation of 21 different chirality SWCNTs.

Sanchez-Valencia and coworkers presented the single-chirality (6,6) and defect-free SWCNTs with lengths up to a few hundred nanometers [57]. A variety of carbon nanorings were created by these and other groups after the groups of Jasti, Itami, and Yamago developed acceptable synthetic techniques for the synthesis of cyclo(para-phenelene)s (Figure 11). Recently, a new N-doped zigzag carbon nanobelt design was carried out. Four benzene rings were fused to an N-heterocyclacene in nanobelt, where eight CH units were swapped out for nitrogen atoms. It is suggested that these nanobelts serve as new building blocks for SWCNTs [79,80]. In the same way, polycyclic aromatic hydrocarbons can be used as graphene templates. They are structurally similar to the basic unit of graphene, that is, graphene nanoribbons (GNRs), which then can be used as templates for SWCNTs [81,82].

In spite of the achievements, grown SWCNTs showed a broad diameter distribution and sometimes did not fully reproduce the template shape. This is particularly due to the instability of templates at high temperatures. Therefore, more stable templates still need to be explored.

## 3. Separation of SWCNT

In this part of the review, we will show several methodologies that have been used to sort SWCNT by their electronic type and chirality. The focus of this section is on the discussion of the first and most recent research study as well as the common mechanisms, advantages, and disadvantages underlining all sorting processes. Finally, we will discuss future prospects and the current stage of automatic sorting, which can bring it from laboratories to industry.

### 3.1. Separation by Electronic Type

SWCNTs exhibit a degree of freedom in a multitude of helicities, which results in a mixture of metallic and semiconducting (Figure 12a) and different-diameter SWCNTs in an unsorted sample (Figure 12b). The SWCNTs have distinct colors depending on the chirality (Figure 12c). The SWCNT synthesis methods have lack of sufficient control over the SWCNT structure, leading to the inhomogeneity of the films’ properties, mainly in the electronic type. It becomes important especially for electronic and optoelectronic applications, where a reliable and reproducible performance of type-separated SWCNT is required. Many sorting approaches originate from biotechnology since SWCNTs and biological things are comparable to one another. Therefore, a variety of separation methods, including electrophoresis, gel chromatography, DNA wrapping, and density-gradient ultracentrifugation, have been developed for the separation of SWCNTs by various electronic kinds (DGU).

#### 3.1.1. Electrophoresis/Dielectrophoresis

The most powerful method of SWCNT sorting by an electronic type is electrophoresis (DEP) [84,85,86,87,88,89,90]. Due to their distinct dielectric constants, metallic (m-) and semiconducting (s-) nanotubes can be sorted by the applied current in this manner. It is known as dielectrophoresis if the nonuniform electric field is activated (DEP). Semiconducting tubes with a lower dielectric constant and negative DEP are left in the solvent, whereas metallic nanotubes with a higher dielectric constant and positive DEP are gathered on the electrode for the strong applied electric field (Figure 13a).

Commonly, three types of DEP are used to separate SWCNT [91]: free-solution dielectrophoresis, gel dielectrophoresis, and capillary dielectrophoresis. In contrast to gel DEP, which requires a matrix often consisting of agarose, agar, or polyacrylamide and magnifies the variations in mobilities of various types of SWCNTs, free electrolysis performs separation without the use of a matrix. Finally, the capillary DEP is performed using tiny tubes called capillaries, which have a diameter of less than 1 mm and provide great resolution.

One of the first studies of separation of SWCNT by free DEP was performed by Krupke et al. [85]. In this investigation, the applied AC field caused the m- and s-SWCNT to travel in different directions along the gradient of the electric field. S-SWCNTs exit the solvent, whereas m-SWCNTs are drawn to the microelectrode array. By adopting an H-shaped microfluidic channel, Shin et al. created this technology and produced a novel method for SWCNT separation, enabling the scalable separation of SWCNT [88].

Kim et al. created p-hydroxybenzene diazonium salt through covalent functionalization and separated it using free-solution electrophoresis. Due to the functionalization’s increased SWCNT mobility in liquid, a negative charge was soon induced on the metallic SWCNTs, allowing for later separation [89,92]. The metallic CNTs can then be recovered by thermally annealing the surfactant for 1 h at 300 °C. It was also reported that free-solution DEP separation of SWCNTs can be achieved by dispersing with a nonionic surfactant, polyoxyethylene stearyl. Inherent zeta potential discrepancies between s- and m-SWCNT were viewed as the cause of this divergence.

However, the above-mentioned articles did not show the high-purity s-SWCNT sorting. A further improvement of this method was achieved by Ihara et al. The SWCNT solution was subjected to a vertical electric field using the vertical electrophoretic cells. Using the plan shown in Figure 13a, different layers with unique horizontal borders formed in the cell after many hours of voltage application (Figure 13b).

In gel DEP, under the applied field, the gel medium sorts analytes by differences in molecular weight or length. In [86], the negatively charged SDS-dispersed SWCNTs were migrated to the anode after the application of a direct-current (DC) voltage through a glass tube (20–40 min). They showed pure s-SWCNTs separated by gel DEP, which contrasted with the free-solution DEP. In addition to the purity, the separation duration was shortened to approximately 20 min, and high-yield sorting was achieved (Figure 13c). Additionally, several gels (agarose, polyacrylamide, and starch) and surfactants (sodium dodecyl sulfate, sodium dodecylbenzene sulfonate, sodium cholate, and sodium deoxycholate) were examined. The most effective combination, however, is SDS and agarose.

A dielectric medium, such as agarose gel, can be used to scale up electrophoretic processes. If both metallic and semiconducting SWCNTs are present in the suspension as individual nanotubes, complete separation can be achieved. Moreover, this method cannot be adopted to sort SWCNTs by chirality in comparison to other sorting methods. As a disadvantage, the used chemicals (e.g., agarose gels) are difficult for postprocessing removal. However, this method is universal and can be applied to either small or large quantities of nanotubes. The highest level of electronic-type sorting that has been achieved up to date is 98%. This limits the adoption of electrophoretic methods in the industry in comparison with other postprocessing sorting methods.

#### 3.1.2. Density Gradient Centrifugation (DGC)

An important advancement in the postprocessing of SWNT materials came with the development of density gradient centrifugation (DGC) techniques. The SWCNTs are propelled toward their isopycnic sites in the presence of a centrifugal field, where the buoyant density coincides with the local gradient density. Surfactant-produced micelles also aid in regulating density. The density-sorted layer is extracted using well-established fractionation procedures after the SWCNs have layered in the gradient in accordance with their buoyant densities.

Centrifugation can be classified as low speed (<10,000 rpm) (also called normal speed), high speed (up to 21,000 rpm, 120,000× *g*), and ultra-speed (>30,000 rpm, >600,000× *g*). Commonly, three types of gradients are used to separate SWCNT: linear, steplike, and nonlinear. In conventional linear density gradients, identical layers with different densities contain similar density species that overlap, and nanotube separation resolution is essentially lost. Nonlinear gradients that contain different height layers with different densities are designed to resolve targeted SWCNT species due to the shortening of their traveling distance in gradient medium.

#### 3.1.3. Low-Speed DGC

By using low-speed centrifugation, Tanaka et al. established a quick and scalable approach for separating metallic and semiconducting SWCNTs [93]. This method is based on the selective interaction of SDS-dispersed m- and s-SWCNTs with an agarose gel matrix. While s-SWCNTs are absorbed by the agarose gel matrix, m-SWCNTs remain in a free state as SDS micelles in the interstitial solution of the gel. In contrast to metallic nanotubes, which remained in the free state with SDS micelles in the solution after centrifugation, s-SWCNTs were selectively retained in the gel. Additionally, it is discovered that by maximizing the gel concentration and agarose composition, the purity of the m- and s-SWCNTs obtained by centrifugation may be increased. Moreover, only SDS with agarose showed the best separation performance, which indicates the importance of surfactant and gel combination selections. This result also indicated that in previous studies of SWCNT electrophoresis separation based on agarose, the applied electric field was not critical and was used only to speed up the separation process. A weak field centrifugation was also found effective for SWCNT separation [94]. To distribute SWCNT in this study, comb polyarylether (PAE) polymers with a strong amphiphilic property were used. This causes heavy liquids to have very low viscosities, which results in very high levels of semiconductor enrichment when employing solely traditional centrifugation.

#### 3.1.4. Ultrahigh DGC

In 2003, Z. H. Chen showed the first attempt to separate m-SWCNTs from s-SWCNTs by density gradient ultracentrifugation (DGU) from surfactant-dispersed SWCNTs containing bromine [95]. Bromine preferentially adsorbs on m-SWCNTs, allowing m- and s-SWCNTs to be separated by centrifugation, according to both density functional theory (DFT) and tests. Similarly, the preferential interaction with halogen elements was reported by Voggu et al. [96]. Moreover, they showed that tetrathiafulvalen electron donor molecules transform the remaining semiconducting SWNTs in an enriched metallic solution to metallic ones through charge transfer. Octadecylamine physisorption on s-SWCNT sidewalls is also used for SWCNT by electronic type [97].

However, in 2006, Hersam and his group boosted the separation of SWCNTs by DGU showing high-level sorting. Different electronic structures of SWCNTs interact with particular surfactant combinations differently during DGU, resulting in the creation of two separate layers of metallic and semiconducting nanotubes, respectively (Figure 14). In [98], it was demonstrated that distinct metal–semiconductor separation occurs when surfactant concentrations of SDS and SC are equal. By tuning the cosurfactant ratio, the significantly improved isolation of only metallic (3:2 SDS/SC) or only semiconducting (1:4 SDS/SC) SWNTs was observed. Later on, Green et al. obtained monodisperse metallic SWCNTs with angstrom-level control over SWNT diameter highly suitable for optoelectronic applications [99]. Later research by Antaris et al. showed that the X-shaped block copolymer Tetronic 1107 and the linear block copolymer Pluronic F68 may be used to separate metallic and semiconducting nanotubes, respectively [100]. Following Arnold et al.’s success, Blackburn and coworkers tuned the ration between metallic and semiconducting nanotubes in films precisely varying the ratio between surfactants [101].

In most of the above-mentioned studies, iodixanol was used as a gradient medium. Such medium is expensive and also contains iodine atoms, which can also dope SWCNTs. Metal–semiconductor separations using sucrose as a gradient medium was shown in [102]. This study is the first report on metal–semiconductor separation using a gradient medium other than iodixanol. The variation between SDS and SC surfactant results in tuning the ratio between metallic and semiconducting components. Moreover, sucrose can be easily removed from the from the sorted CNT material. Niyogi et al. showed that the complicated cosurfactant approach can be replaced by a simple approach based on the SDS interfacial behavior at the nanotube surface in the presence of added electrolyte (alkali salts). Moreover, in this study, the ultracentrifugation time was reduced to 6 h in contrast to previous studies [103]. Hároz et al. used the combined approaches presented in [103] and successfully separated metallic nanotubes from an as-prepared mixed SWCNT solution [99]. Then they used cosurfactant approach mixing SDS, SC, and DOC [102].

Only micrograms of sorted nanotubes are described in each of these studies. The size of the centrifuge is the sole constraint on the method, though. The same methods can be used to sort gram quantities more quickly using a larger-capacity industrial centrifuge. A high-purity sorting up to 99% was achieved using this method to an industrial-scale sorting of SWCNT. However, most of the sorting techniques are applicable only for relatively small-diameter SWCNTs. Meanwhile, the commercialization of the DGC approach has already been initiated by NanoIntegris.

#### 3.1.5. Chromatography

Chromatography is a standard analytical method for SWCNT separation on a wide scale. SWCNTs dispersed in the solution can be sorted using a variety of chromatography techniques, including size-exclusion chromatography (SEC), anion exchange chromatography (IEC), and electrokinetic chromatography, due to the diversity of the stationary and mobile phases.

The most commonly used gel chromatography approach for separation SWCNT by electronic type is column gel chromatography. The most successful separation technique that has been reported is based on polymeric gel beads (stationary phase). Shortly, columns filled with beads of different sizes and shapes can be used for separation (Figure 15). Therefore, a fraction of SWCNT can penetrate through gel, leaving the majority of the SWCNTs trapped. Moreover, some surfactants can provide control of the interaction between SWCNTs and gel beads.

One of the first separation methods based on agarose gel beads in a column was reported by Tanaka et al. (Figure 16b) [86,93]. The SDS surfactant results in adsorption SDS-stabilized s-SWCNTs onto the stationary phase, and therefore, the m-SWCNTs were extracted. At the second step, the SDS surfactant was replaced by DOC, which resulted in the desorption of s-SWCNTs. It was shown that the time duration of the chromatography can be controlled by the size of the beads. Following Tanaka’s work, Liu et al. reported a repeatable single-surfactant multicolumn gel chromatography [104]. This method allows for separating m-SWCNT and s-SWCNT with a different chirality using only SDS. Later on, Liu et al. improved the above-described gel-based separation and used temperature to control the absorbability of SDS-wrapped SWCNTs onto the dextran-based gel [49].

Dispersant removal from isolated SWCNT solutions remains difficult despite significant field development. Additionally, the majority of sorting techniques only perform well with SWCNT of tiny diameter. Gel chromatography and iterative processing are compatible, which has enabled electronic-type enrichment purity to reach 99.9%. This technique has exhibited good yields and purity levels for large-volume electronic-type sorting, enabling industrial-scale sorting.

#### 3.1.6. Two-Phase Separation

Phase separation and mixing of various phase components are the two main phases in two-phase separation (Figure 16a). The final phase can take a few seconds to several hours. As a result, centrifugation is frequently employed to hasten the separation. Commonly, two water-soluble polymers are employed in two-phase separation. Reaching a critical concentration, they form separate layers of different hydrophilicity/hydrophobicity. By adjusting the concentrations of the two polymers, it is possible to fine-tune the difference in hydrophobicity between the two phases. The thermodynamic separation of SWCNTs into two phases takes place. Commonly, polyethylene glycol (PEG) and dextran (DEX) are used as polymers, while a combination of SC or DOC and SDS as surfactants. Khripin et al. first reported the successful separation of CNTs according to diameter and electrical character using this method (Figure 16b,c) [105]. They discovered that semiconducting tubes were formed in the more hydrophobic PEG-rich phase, while metallic tubes were layered in the more hydrophilic dextran-rich phase.

The great progress in the separation SWCNTs by electronic type was achieved by Fagan et al. [105]. In this work, they separated SWCNTs not only by diameter but also by chirality involved the usage of SWCNTs dispersed in a combination of surfactants, such as SDS, DOC and SC. Multiple single species can be isolated in approximately six steps. They successfully sorted metallic (5,5), (6,6), and (7,4) species; (7,4) is nonarmchair metallic not previously isolated by other techniques. Furthermore, they sorted semiconducting nanotubes with a different chirality (e.g., (6,5), (8,3)), and a fraction of (7,7) and (8,5) was observed, but has not yet been isolated. This work was similar to Subbaiyan’s [106], in which the separation steps were reduced from six to eight to just two. The polymers used by Ao et al. in their study in addition to PEG and DEX include PVP, polyacrylamine (PAM), and polyethylene glycol diamine (PEG–DA). According to reports, PVP is positively charged and can interact with SWCNTs that have DNA on them [107]. PEG–DA is positively charged due to the presence of an amine group that induces an electrostatic-based separation mechanism. Following this work, the separation of metallic, quasi-metallic, and semiconducting SWCNTs using DNA-assisted two-phase separation was presented [83]. The chosen species is divided into the top phase of either a PEG/PAM or PEG/DX system in each of the three scenarios, and is then separated from the remainder of the population in the bottom phase. One of the works performed by Eremina et al. showed that a high-purity semiconducting SWCNT fraction can be achieved either by tuning the polymer–surfactant concentrations or by temperature control proved by optical absorption measurements [55] and PL measurements [108]. Recently, Ouyang et al. showed the enrichment of semiconducting nanotubes in a solution (99.9% semiconducting purity), which was proved by the device performance [38]. A novel research study showed a production scale quantity of sorted SWCNTs precisely controlled by temperature [51].

Two-phase sorting can achieve purities above 99.5%. It is highly suitable for industrial production due to the cheap polymer involved, scalability, and short sorting time. The approach is effective because it can increase the diversity of even the smallest populations of species in a sample. Moreover, it can be easily tuned to achieve different ratios of metal and semiconducting nanotubes in the sample. One of the biggest disadvantages is the removal of polymers used, even after several rounds of washing and temperature treatments.

#### 3.1.7. Selective Solubilization and Selective Reaction

SWCNTs are not solubilized in any solvents; therefore, different surfactants are successfully used to solve this problem. Moreover, some solubilization agents are sensitive to the electronic properties of SWCNTs. It was previously demonstrated in this review paper by Tanaka et al. that SWCNTs isolated by sodium dodecyl sulfate (SDS) were embedded in agarose gel, demonstrating that electrophoresis, centrifugation, and mechanical techniques are not responsible in separation. While s-SWCNT are absorbed by an agarose gel matrix, m-SWCNTs remain in free state as SDS micelles in the interstitial solution of the gel. Further electrical and centrifugation techniques just speed up the separation process.

The adsorbed amines are removable after separation, and metallic SWNTs are more firmly adsorbed by amines than semiconducting SWNTs. The theoretical and experimental works performed by Maeda et al. show the selective interaction between methylamine and metallic SWNTs, which results in highly enriched m-SWCNT solutions [109]. The mechanism of this type of separation is described by Lu et al. [110]. Octadecylamine physisorption on s-SWCNT sidewalls is also used for SWCNT separation by electronic type [97].

The selective interactions of porphyrins with s-SWCNT was shown Li et al. [111]. One argument is that a conjugated macromolecule more closely resembles a semiconducting SWNT, and that the surface characteristics of the nanotube are conceptually comparable to those of radical ion pairs, making them susceptible to interactions with free-base porphyrin molecules. Following this work, Wang and coauthors modified the protocol, adding 1-docosyloxymethyl pyrene as the planar aromatic agent. The results were evaluated by enhancing the electrical conductivity of the prepared device.

Among interactions with chemicals, SWCNT can be sorted by physical methods, such as an interaction with electrical current, plasma, laser, microwave, and even special scotch tape. One of the methods for a selective reaction is based on devices that connect source-drain electrodes with semiconducting and metallic tubes. The metallic tubes can be burned off by escalating the drain-source voltage to a high-enough potential when there is oxygen present [112]. M-SWCNTs are more sensitive to light irradiation than their semiconducting counterparts, so preferential oxidation of metallic carbon nanotubes can be achieved. It is feasible to burn off the metallic tubes in a manner akin to the one previously described. H_2_O plasma has been used to etch away the amorphous carbon without causing significant change on the micropatterned arrangement of nanotubes. A similar method based on selecting semiconducting over metallic nanotubes based on a dry etching process of hydrogen plasma was presented in [113]. Because metallic nanotubes’ C–C bonds are simpler to modify than semiconducting ones, the procedure may have a more selective interaction. Song et al. successfully removed metallic SWNTs using microwave radiation [114]. After being heated for a few minutes in a 2.5 GHz microwave, metallic nanotubes burned more quickly than semiconducting nanotubes. Another interesting separation technique is a selective interaction with chemically modified soft polydimethylsiloxane (PDMS) thin films, which act as “scotch tape”. The phenyl-functionalized P-scotch tape specifically targeted m-SWNTs, while the amine-functionalized A-scotch tape eliminated s-SWNTs with selectivity. This approach works best with individual widely scattered nanotubes.

### 3.2. Separation by Single Chirality

The sorting of SWCNTs by chirality has been an aim for two decades. It is essential for future progress in applications in the field of electronic and optic devices. In this section of the study, we give a current summary of the significant sorting advancements on the chirality-controlled synthesis of SWCNTs. We will focus on the techniques that have used various approaches, including density gradient ultracentrifugation, chromatography, and two-phase separation.

#### 3.2.1. Density Gradient Centrifugation (DGC)

One of the first well-known articles about DGC of CNT was published by O’Connell and coauthors [115]. The SDS dispersed in heavy water SWCNT was centrifuged in the solution. It demonstrated how bundles of nanotubes interact with metallic tubes to significantly expand the absorption spectra and extinguish fluorescence. This work demonstrates the individual chirality nanotubes’ strong fluorescence.

Since this study, the most known works on DGU belong to the Hersam group [98,116]. At that time, it became one of the most widely used techniques for the sorting of SWCNTs by diameter/chirality. A clear colored fraction of single-chirality SWCNT was obtained. It was discovered that the density of SWCNTs increased as the nanotube diameter increased (Figure 17). Sodium cholate (SC) or even cosurfactants were later added to improve the fine sorting by chirality, whereas the DNA-wrapped SWCNTs were initially utilized in Arnold’s investigations.

Ghosh et al. further developed this method and showed an impressive single-step separation using nonlinear density gradients [66]. They resolved 10 different (n,m) species: from top to bottom of the centrifuge tube, the bands were enriched by (6,4), (7,3), (6,5), (8,3), (7,5), and (7,6). It was verified by distinct colors of separated layers: (6,5) band was purple in color, (8,3) and (7,6) were green, and (7,5) was blue and, therefore, had distinct absorption and emission peaks in a visible region [117]. In 2010, Zhao et al. used the cosurfactants of SDS and DOS and fine-tuned the SWCNT sorting and obtained (6,5) SWCNTs with ultrahigh purity (97%). While the majority of separation techniques were focused on narrow-diameter nanotubes, Yanagi et al. reported the separation of different-diameter high-purity m-SWCNTs using cosurfactants [102].

The existence of filling within specific CNTs is another crucial factor that can have a big impact on the sorting outcome. Sofie Cambré et al. showed that empty (end-capped) and water-filled (open) carbon nanotubes, which coexist in aqueous solutions, can be separated by DGU [118]. The water-filled and empty SWCNTs showed different diameter–density relations, and therefore, two distinct bands of the same chirality SWCNT were observed. The influence of silver chloride filling on optical properties of metallicity-sorted SWCNT is shown by Kharlamova et al. [119,120]. A recent article published by H. Li and coauthors showed endohedral filling with perfluorooctane and sorting the filled SWCNT. The successful filling by perfluorooctane was confirmed by a detailed comparison with empty and water-filled SWCNTs. The significant changes in the absorption intensity and its peak position were observed as a change in the Raman modes [121].

High-purity sorting up to monochiral purities above 99% was achieved using one-step DGC. The progress reported in this part of the review showed selectivity to the range of SWCNTs with a different chirality (n,m), along with scalability, which could make specific nanotube types widely available. Despite all the discussed advantages, this method is time-consuming. It also requires a careful calculation of the gradient density for the tuning of chirality.

#### 3.2.2. Chromatography

To achieve scalable chirality separation, the various gel chromatography columns are joined vertically in series. Thirteen major chiralities may be differentiated, thanks to the separation. It was discovered that nanotubes of small diameters were initially adsorbed on the gel, indicating a stronger affinity to the medium [104,122]. The control of diameter and chirality was achieved by the choice of surfactant composition. This method allows the mass production of sorted SWCNTs. In addition, it requires less time and is less expensive in comparison with DGC. The next step was fine-tuning the SWCNT chiral angle, thanks to the temperature control. Liu et al. achieved a single-chirality separation of seven SWCNT species using temperature-controlled gel chromatography. In this case, temperature allows the control of the interaction between SDS-wrapped SWCNTs and the allyl dextran-based Sephacryl gel [123]. Flavel et al. showed that Sephacryl gel columns can be reduced to one by altering the pH of the SDS. In this study, the separation of 15 different nanotube (n,m) species with a purity of 17–72% and then 8 with a purity of 61–95% was demonstrated [124,125]. The chirality selectivity between the gel and the surfactant-wrapped SWCNTs was given by the mixed-surfactant system made up of SDS, SC, and DOC. In particular, chiral-angle selectivity and diameter selectivity were achieved in the SC/SDS and SC/SDS/DOC systems, respectively [126]. In this article, a scalable, simple, and fast process enabled the milligram-scale separation of single-chirality (9,4) and (10,3) SWCNTs with high purity (>90%) and high yield.

The technique is also very adaptable and may be quickly expanded to extract SWCNTs with different chiralities. To satisfy the needs of the industrial application, the process can be simply scaled up. However, this method requires multiple steps for high purity.

#### 3.2.3. Two-Phase Separation

The selectivity of specific SWCNTs to SDS and DOS allowed for separating them by using two-phase separation (Figure 18a). It was described first by Forgan and coworkers. In this article, the separation of specific small-diameter SWCNT species was observed in a PEG–DEX complex. The smaller-diameter CNTs were present in the more hydrophilic dextran-rich phase, while larger diameter (7,5) and (8,4) tubes were found in the more hydrophobic PEG-rich phase [127]. In [105], it was demonstrated that this effect highly depends on the curvature of the SWCNT, which makes smaller-diameter CNTs, in the so-called smaller-diameter regime, less hydrophobic. Interestingly, in a large-diameter (>1.2 nm) regime, a nanotube’s polarizability renders semiconducting tubes more hydrophobic than metallic ones.

It was also found that the diameter distribution in the sample can be tuned by the ratio between SDS and SC [129]. A previously constructed theoretical model is used to explain these variations in separation efficiency with surfactant composition and predicts that surfactant mixes modify the charge per unit length of particular SWNTs, changing the separation. Further tuning of the chirality can be achieved by temperature control. A high temperature weakens the interaction between the surfactant and CNT, changing the ratio between fractions of separated SWCNTs. To add more of the desirable species to a top phase or to remove an unwanted species, little cooling or heating can be applied [130]. Another way to control the separation efficiency is redox reaction, which can enhance the isolation of certain SWCNTs in the top or bottom phase. Using all the parameters, temperature, surfactant ratio, and reaction efficiency, the two-step separation can be achieved [106]. The course of separation can be further controlled by switching a PEG–DEX system to a more hydrophilicity/hydrophobicity system. In addition, DNA wrapping improves the separation efficiency. Overall, the virgin combination has been successfully purified into 15 single-chirality nanotube species.

It is important to note that the two-phase separation is one of the techniques that can be used to separate larger-diameter CNTs. Recently, using pH-driven sorting, 11 different (n,m) species were isolated in a simple one to three steps (Figure 18b). In this article, a universal recipe to separate smaller and larger diameters was presented [128].

Due to its strong chirality selectivity and extensive coverage of diameter separation, two-phase separation is deemed one of the prospective approaches suitable for future commercial applications. However, used polymers have to be removed for further applications, which become a nontrivial problem.

## 4. Applications of Type and Chirality-Separated SWCNT

The effect of varying the ratio of metallic and semiconducting tubes with different bandgaps on optoelectronic properties was widely used for a wide range of applications (Figure 19). For several application sectors, selective enrichment of CNTs with particular electrical characteristics is necessary. Semiconducting tubes are frequently doped (ambient doping); therefore, both kinds of tubes, metallic and semiconducting, can carry current [131]. This becomes critical to interconnect applications. The inclusion of metallic CNTs in field-effect transistors (FETs) can cause device failure. While metallic SWCNTs provide excellent conductance throughout the electrodes, CNT’s semiconducting characteristics are crucial for collecting and transferring electrons in perovskite solar cells. Therefore, the ratio between metallic and semiconducting nanotubes becomes critical. There is also a growing interest in utilizing carbon nanotubes for a variety of biomedical applications that take advantage of the optical properties of single-chirality CNTs. In this part of the review, we will discuss the progress of using type and chirality-separated nanotubes in a wide range of applications. This section describes the groundbreaking studies and improvements that have been made in the use of sorted nanotubes with clearly specified properties in a variety of scientific domains.

### 4.1. Field Effect Transistors (FETs)

One of the methods commonly used to excess the enrichment degree of CNT separation is FET. SWCNTs are desirable channel materials for a variety of field-effect transistors because of their high degree of purity. This is caused by the high current-carrying capacity and charge transport mobility. This is an important parameter for the FET and the on/off ratio, which is the ratio of the “on” state conductance when the gate is open to the “off” state conductance when the gate is closed. In this case, the metallic nanotubes cannot be controlled by the gate, and furthermore, being present in the percolation network bridges the FET channel. The highest switching ratio is currently at 2 × 10^8^ for pure semiconducting SWCNTs [132], while for mixed CNT, it does not reach values higher than 10^3^–10^5^. Before the separation techniques were obtained, a number of FETs were designed and tested [133,134,135,136]. One of the first FETs based on sorted metallic and semiconducting nanotubes was shown by Arnold and coworkers [98]. The high separation performance was confirmed by the significantly different behavior of FET.

Most widely used semiconducting materials, including Ge, InAs, and InSb CNTs, have bandgaps below 1 eV and carrier effective masses that are significantly lower than those in Si. Due to these characteristics, they are susceptible to ambipolar behavior, especially when the drain voltage bias is strong and the off state is extremely pure. Low-standby-power dissipation electronics started to depend on this. A low bandgap caused by a small potential barrier near the drain in the off state is maybe SWCNT’s worst drawback. Suppressing off current in CNT-film-based FETs remains a big challenge.

### 4.2. Sensors

SWNT-based FET has been used to sense the change in the temperature and humidity as well as various chemical and biological species. This is due to the high physisorption, chemisorption, high aspect ratio, and sensitivity of electrical transport to temperature and humidity [137,138,139,140]. The capacitance of the CNT-based sensors decreases with increasing humidity (Figure 20) [141]. The exponential recovery after the annealing treatment proved a high level of performance accompanied by a fast response time of 2–8 s. The electrical transport of CNTs also depends on the temperature and can be used for temperature sensing [140]. The sorting can further improve the device’s performance.

Ganzhorn et al. found that FET-based s-SWCNTs are extremely sensitive to hydrogen, with a strong dependence on the nanotube diameter [142]. They discovered, in particular, that small-diameter s-SWCNT devices had reduced sensitivity, which was accounted for by a change in the nanotube work function caused by the chemisorption of “spilled-over” atomic hydrogen from the Pd electrode, which compensated for the changes in the Pd work function. In contrast, medium-diameter s-SWCNT demonstrated greater performance even in ambient air, regardless of the substrate employed. Sorted (16,2) s-SWCNT demonstrated excellent NO_2_ sensing capabilities with high mobility and on/off ratios, minimal hysteresis, and low operating voltage [143]. Additionally, at room temperature, NO_2_ sensors based on the sorted s-SWCNTs exhibited great sensitivity and stability, a quick response time of 30 s, and a low limit of detection. It was demonstrated that employing s-SWCNTs significantly increased the sensitivity to NH_3_. Using s-SWCNT films, the sensitivity to NH_3_ was increased 2.5 times, enabling the creation of selective sensors based on the chirality of the SWCNTs [144]. The charge transfer-mediated physisorption process in the nitroxide–CNT system directly depends on the metallic nature of the samples because it is twice as strong for metallic tubes as for semiconducting ones [145]. In addition, the sensor performance highly depends on the percolation of m-SWCNT in an enriched SC-SWCNTs network. Sensing is dominated by the modulation of the Schottky barrier below the m-SWCNT percolation threshold, but above this threshold, it is only attributable to a charge transfer between SWCNT and gas molecules [146]. Ammonia gas was detected at room temperature using PF8-DPP-sorted sc-SWCNTs as a sensing material. The sorted sc-SWCNT-based ammonia sensors demonstrated great sensitivity (R/Ro 54.4%), quick response (30 s), and good stability when exposed to 0.6% NH_3_ at ambient temperature [86]. A new direction in SWCNT sensing was opened after research about conductance change after the change of pH, which then was studied to detect enzymes [147]. Regarding direct electron transfer as a detecting technique for electrochemical enzyme biosensors, s-SWNTs were shown to be more suited than metallic SWNTs [148].

CNT-based sensors have shown great promise and are being used in many different sensing-related fields. However, it has been discovered thus far that SWCNTs improve ionization in contrast to resistance sensors, enabling the detection of gas molecules with low adsorption energy. Specific functionalization can be used to expand the applications for sensors. The stability and repeatability of CNT-based sensors depend critically on the characteristics and behavior of the sensors, which might vary dramatically depending on the purity of SWCNTs. There is still a paucity of knowledge regarding device degradation for commercial implementation. Slow response and recovery, which are brought on by SWCNTs’ slow gas adsorption and desorption processes, is another drawback that limits the widespread adoption of SWCNT sensors.

### 4.3. Light Emitters and Photodetectors

When the electric current passes through CNTs, the near-infrared and infrared emission is observed. Therefore, SWCNTs have been highly investigated as light emitters. The sorting of nanotubes by chirality gave much narrower emission peaks at the desired wavelength. These emitters exhibit ultrahigh response speed comparable to laser diodes [149]. In addition, for all semiconducting SWCNTs, the excitonic transitions E_11_ and E_22_ locate in the infrared spectral region, important for telecommunications [150,151,152]. To distinguish pure large diameter s-SWNTs with a fundamental transition centered at 1550 nm and strong PL excitonic emission, Sarti and coworkers used PFH-A. Electrically driven monochiral SWCNT light emitters have been shown by a number of groups in a variety of applications, including polarized light emission, narrow linewidth emission [153], trion-based emission [154]. In photodetectors, SWCNTs have a number of benefits, such as room-temperature operation, fast response times, and wide bandgap tunability using SWCNT sorting. It has been shown that a SWCNT photodetector made of highly pure semiconducting SWCNTs can function at room temperature and zero bias. With a broadband response (785–2100 nm) and strong detectivity, 22,500 photodetectors were created on a Si substrate [155]. The narrowing of the full width of half maximum of the detected signals and the improved sensitivity have been achieved by the integration of single-chirality (8,4) and (8,3) SWCNTs into cavities [156]. More impotently, using chirality-specific CNT-film-based signal recognition systems, they developed the concept of the “resonance and off-resonance” cavity, which can be used in a wide range of applications. Despite these developments, the sensitivity of SWCNT-based photodetectors has not reached the single-photon limit as is needed in some sensitive applications, such as quantum information. Huang et al. used high-purity semiconducting CNT and demonstrated a plasmonic electrode structure that could collect photoinduced carriers effectively and enhance light absorption at the same time [157].

The purity of the semiconducting nanotube network is crucial, because metallic nanotubes would simply quench excitons and reduce the efficiency of both IR sources and detectors. Moreover, the exciton diffusion is quenched by nonradiative decay processes (e.g., phonons, defects, tube-to-tube contacts), resulting in low PL quantum yields. Therefore, purified SWCNTs are required for light emitters. In addition, in unsorted SWCNTs due to the energy transfer processes, the excitons transfer to narrower-gap tubes, which also result in the suppression of the emission at a particular wavelength. In this case, monodispersed SWCNTs are required.

### 4.4. Transparent Electrodes

In nearly all devices based on conversion between electrons and photons, electrically conductive SWCNT has to be optically transparent. High optical transmission and low sheet resistance are ideal characteristics for transparent electrodes. Either one of these properties can be tuned to the desired value by changing the thickness of the CNT. In addition, it has been discovered that a number of factors, including the CNT length and diameter, electrical characteristics, chemical doping, purity, synthesis, and preparation procedures, have a significant impact on the resistance and, consequently, conductivity of a CNT network [158,159].

The conductivity of SWCNT films is also dominated by tube-to-tube junctions [101]. In comparison with metallic–metallic junctions, the contact resistance at metallic–semiconducting junctions is three orders of magnitude larger. The “metallization” of semiconducting SWCNT can be achieved by doping. However, the doping effect is known to degrade over time. Another solution is to make transparent electrodes based on sorted SWCNT. Li and colleagues separated a considerable amount of sorted metallic SWNTs [111]. They showed that obtained films had ~80% transmittance (at 550 nm) accompanied by surface resistivity less than 100 Ω/square. This material is already competitive with that of topically used electrode coatings. Rahy et al. created films from separated metallic SWNTs that had an optical transmittance of 80% at 550 nm and showed sheet resistance as low as 130 Ω/square. [160]. I. Jeon and coworkers compared SWNT and graphene as the bottom electrode in inverted perovskite solar cells. Both carbon materials showed high performance slightly bigger for graphene due to the better morphology and high transparency in a visible range. However, due to the entangled arrangement of the nanotube network and their inherently defect-free nature, SWNT-based flexible PSCs demonstrated slightly superior mechanical stability.

It has been also reported that a hybrid structure, such as a combination of AgNWs and CNTs, exhibits superior behavior as transparent electrodes [161,161,162,163,164,165]. In particular, CNT can form conductive bridges between the AgNWs, which enhance the conductivity. While the CNTs filled in the spaces between the nanowires, the AgNW formed a percolation current-carrying network, enabling local electron transport and lowering junction resistance [166,167]. Lee et al. demonstrated the benefits of using both AgNW and CNT on a stretchable substrate and obtained a stretchability of over 460% [168]. The fabricated electrode demonstrated an improved sheet resistance of 62.3 Ω and a transmittance of 83.4% with excellent heating stability, temperature uniformity and resistance to bending [169,170]. Composite films exhibited excellent stability in sheet resistance under the bending tests.

The optoelectronic characteristics of these films must keep becoming better in order for SWCNT transparent electrodes to be widely used in commercial applications. The simplest approach is to increase the DC conductivity of the films. There are numerous approaches, such as creating longer, more pure tubes, SWCNTs with fewer defects, and SWCNTs that are less bundled in the film network. As stated above, a film composed entirely of metallic nanotubes or a mixture of metal and p-type CNTs that have undergone degeneration will have greater conductivity than a film produced entirely of unseparated, undoped tubes, which will considerably enhance the performance of the electrodes. The producing of larger-diameter SWCNTs will also minimize the influence of the residual semiconducting tubes present in metal-enriched SWCNT by lowering their bandgap. However, all sorting methods suffer so far from low yield and high production costs, which lower their applications as transparent electrodes. Additionally, most of the sorting techniques produce monochiral SWCNTs, which have a narrow spectral absorption range, which limits their commercial applications as transparent electrodes.

### 4.5. Photovoltaics/Solar Cells

A transparent, highly conductive electrode is required for a solar cell to produce solar energy. As we showed in the previous section, SWCNTs are ideal candidates for this purpose. They can be implemented either as an extraction layer or as an active layer. Their high conductivity and high work function make them useful for a whole transportation. For a highly effective active layer for perovskite solar cells, Bati and colleagues [171] demonstrated highly enriched semiconducting and metallic SWCNT samples synthesized utilizing a column chromatography approach (Figure 21A,B). The introduction of different SWCNT families with optimized contents resulted in significant enhancements of efficiency. S-SWCNT-based photovoltaic cells operate as a type II heterojunction, usually incorporating an electron acceptor, such as fullerene-C_60_. Bindl et al. showed that exciton dissociation at the s-SWCNT/C60 interface is extremely efficient, approaching 100% [172]. The high charge-separation efficiency was established for sorted (7,6) SWCNTs [173]. Photoelectrochemical studies revealed their ability to harvest light energy into electricity. The trend for charge separation and charge recombination, which demonstrated the best light energy conversion efficiency, was followed by the efficiency of photocurrent generation. We created the first all-carbon solar cell with the rGO anode and doped SWNT cathode, which had high efficiency under VIS and NIR illumination [174]. Additionally, by effectively collecting photogenerated electrons with nanocomposites with high electron mobility, the performance of solar cells can be enhanced. Dang et al. showed that m- and s- SWNTs and their aggregation affect solar devices based on SWNT/TiO_2_ nanocomposites and virus/SWNT complexes [175]. An increased photocurrent was mostly to blame for the performance gain. Lee et al. carried out a comparable work on SWNT/TiO_2_. They then maintained the mechanical and electrical network between the TiO_2_ nanoparticles and SWCNT by adding a tiny number of pure m-SWNTs.

Overall, it was demonstrated that a nanohybrid film based on high-purity (99%) semiconducting SWCNTs produces a number of times higher photocurrent than a film constructed of metallic SWCNTs and unsorted SWCNTs. The Dirac point causes metallic nanotubes to undergo quick Fermi-level relaxation and recombination, which reduces the amount of charge that is transferred to the external circuit. Semiconducting nanotubes, in contrast, provide efficient transport routes for a different charge to reach the active electrode in the photoelectrochemical cell.

### 4.6. Batteries

Typical Li-based batteries consist of a cathode, anode, and conducting electrolyte. In a typical cell, the graphitic carbon anode and lithium metal oxide cathode are employed as the positive and negative terminals of the cell. One of the most promising candidates for the anode material of lithium-ion batteries is CNTs, which are regarded as one-dimensional hosts for the intercalation of Li. Kawasaki et al. confirmed that the reversible Li ion storage capacity of metallic SWCNTs is around five times greater than that of semiconducting SWCNTs by using various techniques to produce metallic and semiconducting SWCNTs [176]. Later on, Jaber-Ansari and colleagues found that metallic SWCNTs accommodate lithium much more efficiently than their semiconducting counterparts. It was also shown that lithium is more easily accommodated at the junctions; therefore, this process is also sensitive to the density of the tubes. Semiconducting SWCNTs, if made denser, also begin to take up lithium at levels that are comparable to metallic SWCNTs [177]. One of the most recent studies showed that mSWCNT-enriched ivylike conductive nanonets were presented as a facile and versatile surface engineering method for battery electrode materials [178]. About five times as many lithium ions can be stored in metallic CNTs than in semiconducting CNTs. Additionally, metallic SWCNT-built batteries reach their full charge a little bit quicker than semiconducting SWCNT-built batteries. Furthermore, distinct Li structures can be seen inside the tubes, indicating that the intercalation by Li ions is significantly dependent on the diameter of the SWCNTs. One of them will increase the lithium capacity: a linear chain in the axis. Additionally, it is noted that the diameter of the nanotubes affects the interaction potential in the core region. CNTs are a better possibility for use as a lithium ion battery anode material because SWCNTs, especially those with smaller diameters, have higher interaction energies. It has been demonstrated that sorted SWCNTs, which have a significant photovoltaic potential, can be used to create all-carbon solar cells.

### 4.7. Neuromorphic Engineering. Quantum Information Processing. Interconnections for Electronic Circuits

By replicating biological neural networks to produce better computation, neuromorphic architectures have emerged as a prominent contender to replace Neumann computing. Recently sorted SWCNTs have also been thought to as neural network simulators [179,180]. In [180], s-SWCNT inks were prepared by using a solution with 99.9% semiconducting purity. The synaptic characteristics of aligned CNT FETs were investigated in this study by the authors, who also offered a thorough characterization based on DC and pulsed observations. They also demonstrated that the learning rate may be optimized by adjusting the CNT synaptic properties. Kim et al. also showed a synaptic transistor based on highly purified, separated 99% semiconducting carbon nanotubes, with reliable, analog, conductance-modulated behavior. They showed a flexible neuromorphic device that has highly desirable properties, such as inexpensiveness, lightweight, disposability, and recyclability.

A key element in developing communication technologies, including quantum information processing, is the single-photon emitter. For this application, a two-level system that emits identical photons one at a time is necessary for the optimum single-photon emitter. This process is possible only when carbon nanotubes consist of defects, which result in defect sites, longer decay lifetimes, and enhanced quantum yield. It was also shown that the integration of dopant states in carbon nanotubes can provide bright and high-purity single-photon emitters on a silicon photonics platform at room temperature [181]. The single-photon emitters at telecom wavelengths (1.3–1.55 µm) were also demonstrated, as well as their ability for integration to existing low-loss optical networks [182]. The chirally enriched SWCNT that can produce the emission at the required wavelength is crucial for these devices. Ma and colleagues further investigated photon antibunching in oxygen-doped (6,5) SWCNTs for this purpose and showed that single-photon emission might be observed in such a configuration [183].

### 4.8. Bioimaging

Numerous methods, such as fluorescence, Raman, photoacoustic, magnetic resonance, and nuclear, can be used for CNT imaging [184]. It is difficult to generate data about organs and isolated cell sites, and small-size probes with steady, brilliant optical signals are needed. Near-infrared (NIR) emission significantly increases light penetration and tissue detection for this purpose. Therefore, bright PL in NIR typical for individualized s-SWCNT can significantly improve the imaging capability. Diao and coworkers used an 808 nm laser resonant to separate (12,1) and (11,3) s-SWCNT, which showed 5.0-fold higher fluorescence brightness compared with unsorted SWCNTs. The biocompatible surfactant-exchanged sorted SWCNT required a dose that was six times lower than that necessary for unsorted SWNTs, which may help allay concerns about the toxicity of nanotubes in vivo. Additionally, the suitable surfactant wrapping can improve the brightness of PL. Sorted (6,5) SWCNTs were wrapped in the biocompatible surfactant C18-PMH-mPEG by Antaris et al., which improved the photoluminescence’s brightness [100,141] (Figure 22). The (6,4) SWCNTs have been used to probe cancer tissues [185].

Despite the advancements sorted SWCNTs have made in biomedical applications, questions about their toxicity remain. The most thorough investigation revealed that, due to the oxidative effects, metallic tubes had a greater impact on bacterial viability and membrane permeability than semiconductor SWCNTs [186]. However, one of the recent works assessed the differences in toxicity of different chiralities of SWCNTs on mammalian cells, bacteria, and the rodent lung [187]. The findings demonstrate that the electronic characteristics and chirality have no bearing on the toxicological effects of SWCNT on biological items.

### 4.9. Conductive Fillers for Composites. Shielding Devices

The metallic SWNTs considerably increase the electrical conductivity of the resultant nanocomposites when distributed in a conductive polymer compared with the original unpurified nanotube sample [188,189]. In particular, they showed that separated metallic nanotubes offer improved performance in a composite based on poly(3-hexylthophene) and other polythiophenes. The conductivity improvement was one order higher than without SWCNTs.

One of the important applications of SWCNT is electromagnetic shielding and modulation [159]. Due to their conducting qualities, metals are generally more suited for EMI shielding than semiconductors. It is interesting to note that films made utilizing metallic tubes only showed a lower conductivity than the s-SWCNT due to the tube–tube junction between nanotubes in the thin film, as demonstrated by Blackburn et al. [101]. Yu et al. demonstrated that the electrical makeup and tube-to-tube junction resistance of the sorted SWCNT are key factors in the electromagnetic shielding performance. A variety of film formulations, including those with 99% m-SWCNT, 99% s-SWCNT, and 99% s-SWCNT wrapped in a polymer, were investigated. Materials with a micron-thick coating of 99% s-SWCNT were found to have the highest EMI SH EF for the sorted SWCNT. It was discovered that the electromagnetic shielding is 35 dB between 8 and 12 GHz and 40 dB between 12 and 19 GHz [190].

## 5. Remaining Challenges and Future Outlook

Each separation technique discussed in this review also has its own unique separation parameters that determine the separation efficiency, purity, cost, speed, and simplicity. The details are described in Table 1. Some chemicals and DNA involved in the sorting of CNTs are expensive on a large scale of sorted CNT production. The cost of the chemicals also results in the improvement of the level of CNT resolution and purity due to the limitation of experiments. For this reason, synthesis, DGC, and ATPE methods can be considered as relatively expensive in comparison with electrophoresis and chromatography techniques. Most of the sorting methods require a better understanding of interaction between polymers and SWCNTs, which is also limited due to the cost of materials. The possible option to solve this problem would be to establish protocols, in which the used chemicals can be recycled. The high costs incurred when sorting CNTs arises from the high cost of the equipment, which slows down laboratory developments. From this point of view, DGC of growth from seeds is rather more expensive than the rest of the techniques. Another factor that makes some techniques more preferable than another is the speed of the sorting. Some of these methods, such as synthesis, DGC, and chromatography, are time-consuming due to the physical processes involved and cannot be improved. Finally, the reproducibility even in industry can be limited by the level of simplicity. Synthesis and DGC require a precise control over the experimental parameters, which increases the complexity of some techniques.

Moreover, the larger production capabilities will significantly improve the quality of the sorting. Techniques such as multistep separation need to be explored. Industry will be interested in the use of new chemicals if they ensure that they can be supplied. In addition, finding new commercially available surfactant systems with high-resolution selectivity is desirable. Furthermore, the sorting method has to be reproducible and scalable.

Chirality-sorted SWCNTs have been used in many advanced applications, such as optoelectronics and nanoelectronics. One of the biggest disadvantages of the mostly used sorting techniques is the presence of chemicals. Only epitaxial growth of SWNTs suffers from this problem. However, the low yield and efficiency caused by the poor activity and thermal stability of the nanocarbon seeds are still the main hurdles.

A huge progress has been achieved in sorting SWCNTs by type and chirality. For some applications, long individualized SWCNTs with a low level of defects are required. However, the successful sorting needs SWCNT unbundling, which follows the preliminary intensive ultrasonication and subsequent centrifugation. In particular, longer exposure to ultrasounds leads to more efficient exfoliation of the CNT bundles and higher functionalization degree. These processes damage the SWCNTs, which results in the deterioration of their properties.

In terms of the challenges among all techniques, the tunability of the sorted diameter SWCNT is far from being reached. Some of the techniques have a lack of the sorting resolution. Most of the sorting techniques are adopted for small-diameter SWCNTs, while for some applications, bigger-diameter SWCNTs are required. Furthermore, the sorting quality has to be improved for the some described applications.

The sorting process may become more appealing to industry with the use of automation. There have been several attempts made in this direction. For instance, adsorption chromatography with hydrophobicity modification can produce a range of single-chirality SWCNTs resolved for 20 chirally enriched SWCNTs. On chromatographic equipment that is available for purchase, SWCNT can be automatically separated. With this technique, it is simple to produce a significant quantity of single-chirality SWCNTs. As a control, the near-infrared optical technologies were suggested. In another example, automatic fractionation was already adopted in the production ultracentrifugation technique.

## 6. Conclusions

In this review, we performed a comprehensive analysis and overview of the status of research on the conductivity and chirality-selective synthesis of SWCNTs and the sorting of SWCNTs. We reviewed the synthesis methods of SWCNTs, such as synthesis from nanocarbon seeds and nanocarbon fragments obtained by organic synthesis and the catalyst-mediated method of SWCNTs. We discussed the methods of sorting SWCNTs by conductivity type and chiral angle. Applications of single-chirality SWCNTs were presented.

## Figures and Tables

**Figure 1 materials-15-05898-f001:**
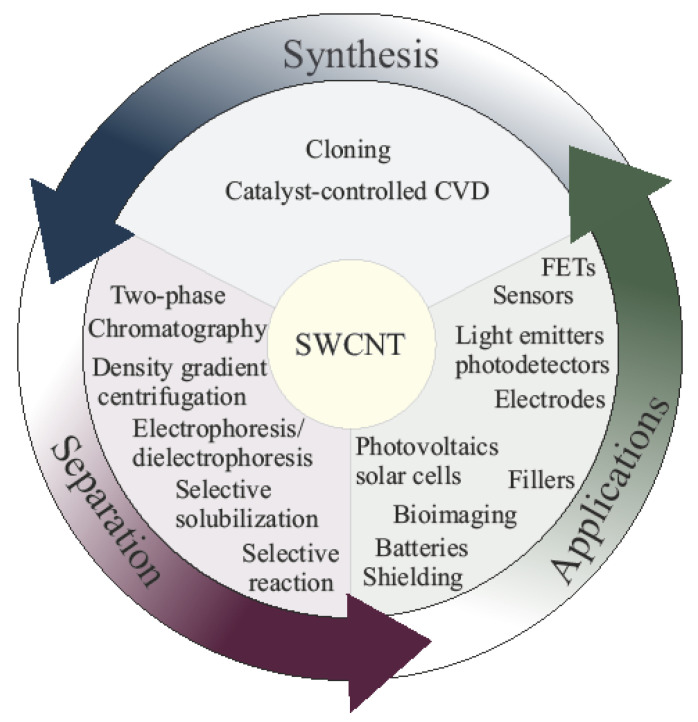
The main synthesis routes, separation methods, and applications of single-chirality SWCNT.

**Figure 2 materials-15-05898-f002:**
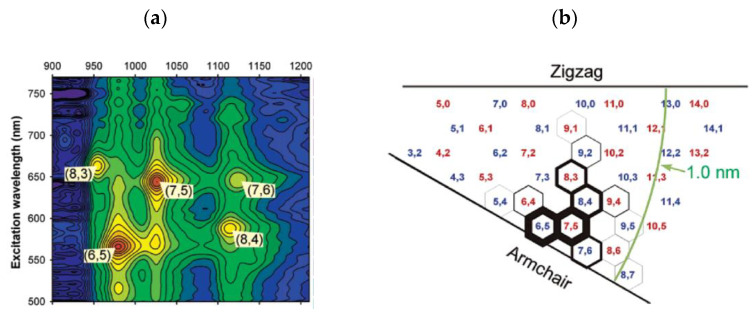
(**a**) Normalized 2D fluorescence map of excitation and emission wavelength of a chirality-enriched SWCNT sample. (**b**) Intensity distribution for all semiconducting (n,m) in the same sample. The edge thickness of the hexagons is proportional to the observed intensity for the (n,m) chirality [46]. Copyright 2010, American Chemical Society.

**Figure 3 materials-15-05898-f003:**
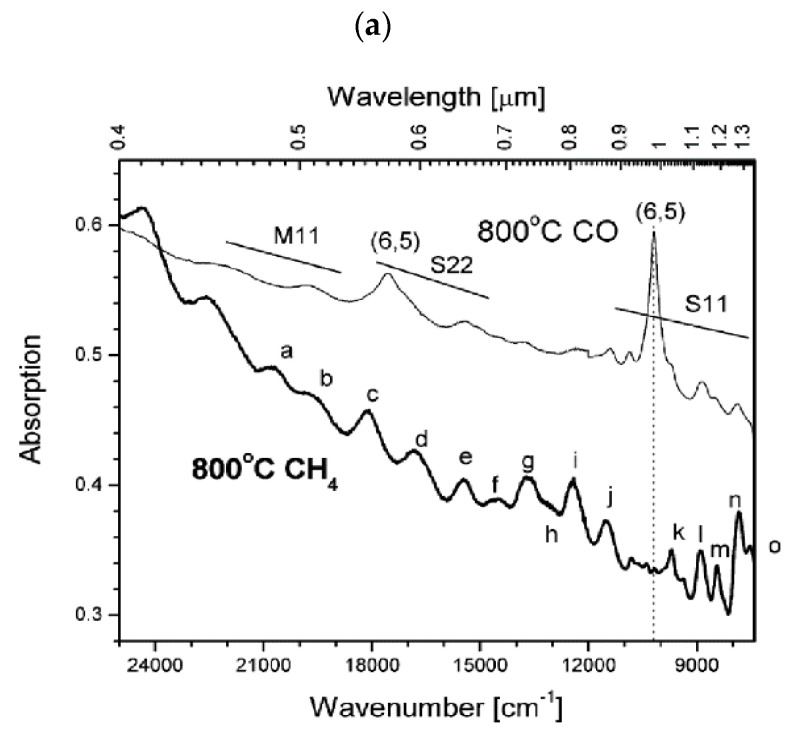
(**a**) Optical absorption spectra of SWCNTs grown from either CO (top) or CH_4_ (bottom) as carbon feedstock. The upper spectrum is dominated by the optical absorption bands of the (6,5) chirality. The lower spectrum signifies a broader nanotube distribution. The individual absorption bands belong to: M_11_ (a, b, and c), S_22_ (d (8,4), e (7,5) (7,6), f (12,2), g (8,6) (8,7), h (10,6), i (12,1), and j (14,0) (11,6) (13,2) (12,4)), and S_11_ (k (7,5), l (8,4) (7,6), m (12,1) (8,6), n (8,7), and o (12,4) (13,2)). (**b**) The Hamada map of SWCNTs is grown from either CO or CH_4_. The diameters for a zigzag and armchair direction are marked at 0, 0.5 and 1.0 nm [39]. Copyright 2006, American Chemical Society.

**Figure 4 materials-15-05898-f004:**
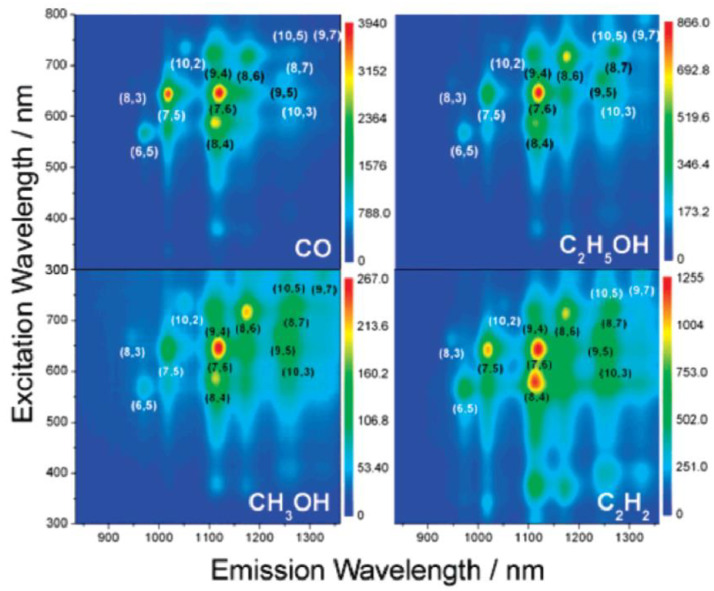
Maps of photoluminescence intensity versus excitation and emission wavelength (in nm) for SWCNTs grown from several feedstocks: CO, C_2_H_5_OH, CH_3_OH, and C_2_H_2_ [38]. Copyright 2007, American Chemical Society.

**Figure 5 materials-15-05898-f005:**
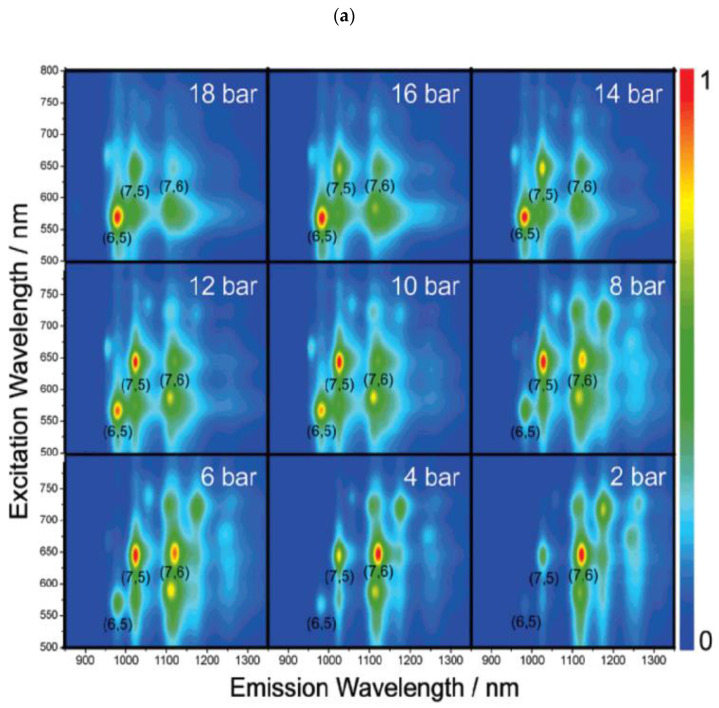
(**a**) PL maps of SWCNT grown under CO pressures between 2 and 18 bar. The labels indicate three dominant chiralities. (**b**) Abundances of SWCNT with different chiralities versus changing CO pressure. (**c**) Optical absorption spectra of SWCNTs obtained under a different CO pressure. The first and second absorption peaks are highlighted, indicating the change in the three main chiralities’ intensities [37]. Copyright 2007, American Chemical Society.

**Figure 6 materials-15-05898-f006:**
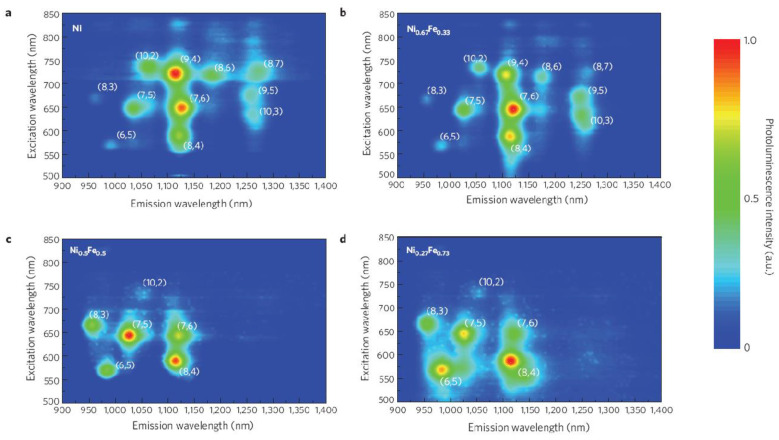
The contour plot maps of the PL intensity obtained for the SWCNT samples synthesized with different catalysts: (**a**) Ni, (**b**) Ni_0.67_Fe_0.33_, (**c**) Ni_0.5_Fe_0.5_, and (**d**) Ni_0.27_Fe_0.73_. The chiral indices are indicated near the corresponding peaks [41].

**Figure 7 materials-15-05898-f007:**
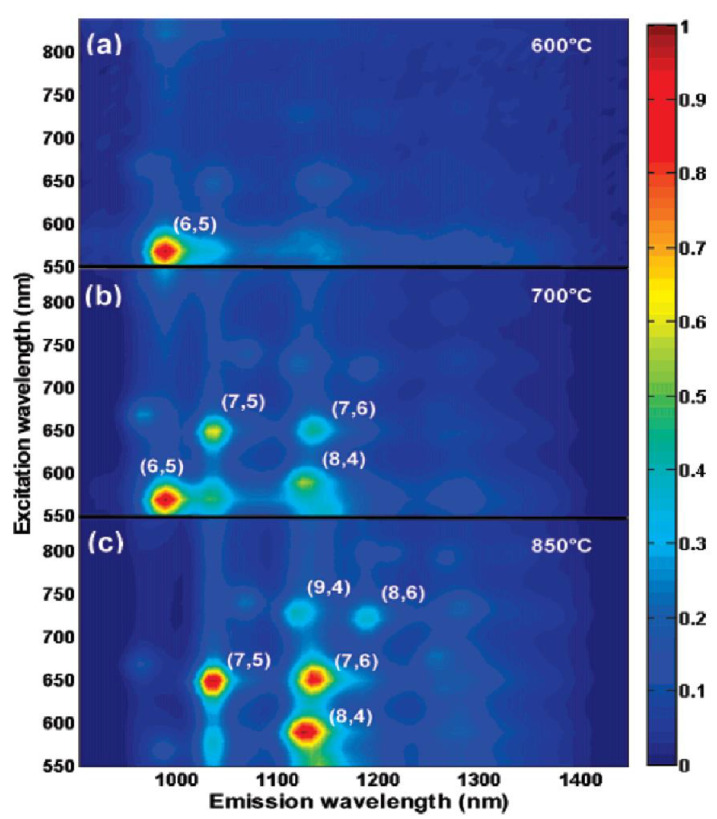
The photoluminescence maps of SWCNTs synthesized at (**a**) 600, (**b**) 700, and (**c**) 800 °C. The chirality indices are labeled near every pronounced peak [58]. Copyright 2007, American Chemical Society.

**Figure 8 materials-15-05898-f008:**
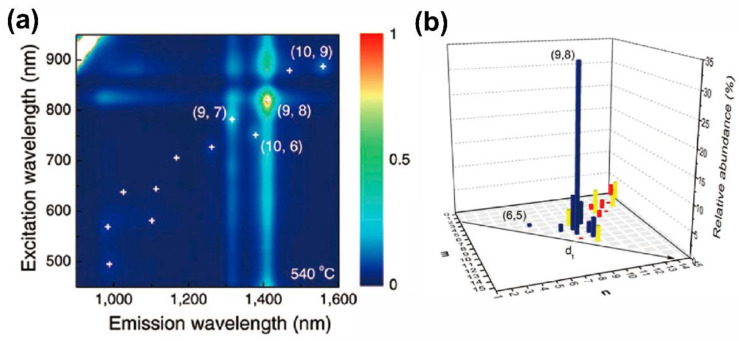
(**a**) PL intensity spectra of SWCNTs dispersed in 2 wt% sodium dodecyl benzene sulfonate (SDBS) D_2_O solution. (**b**) Abundance of (n,m) SWCNTs produced after CoSO_4_/SiO_2_ catalyst reduction at 540 °C using PL (blue), Raman (red), and absorption (yellow) [59]. Copyright 2013, American Chemical Society.

**Figure 9 materials-15-05898-f009:**
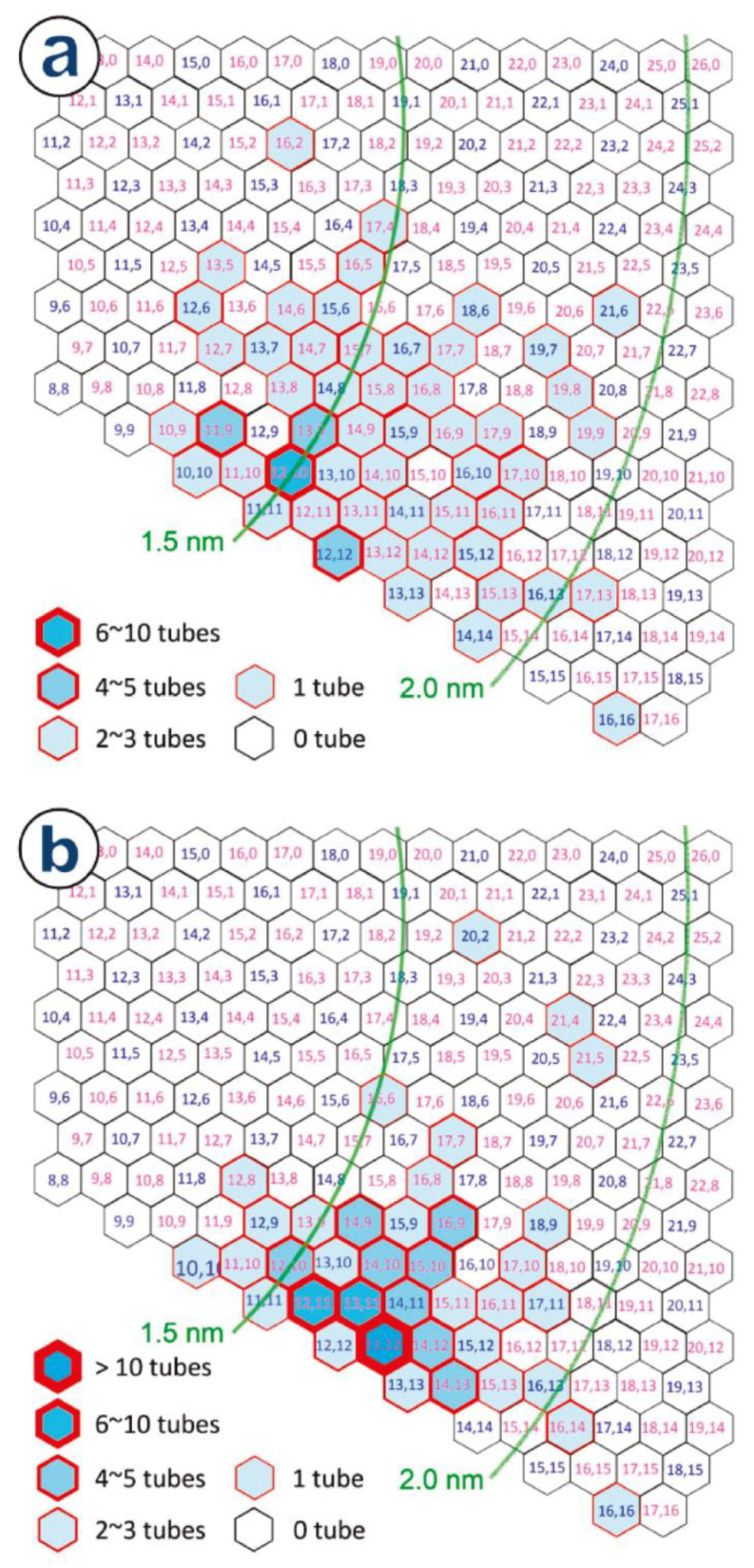
Chirality map distribution of samples produced using an aerosol floating-catalyst CVD process with (**a**) 0 ppm NH_3_ and (**b**) 500 ppm NH_3_ [49]. Copyright 2011, American Chemical Society.

**Figure 11 materials-15-05898-f011:**
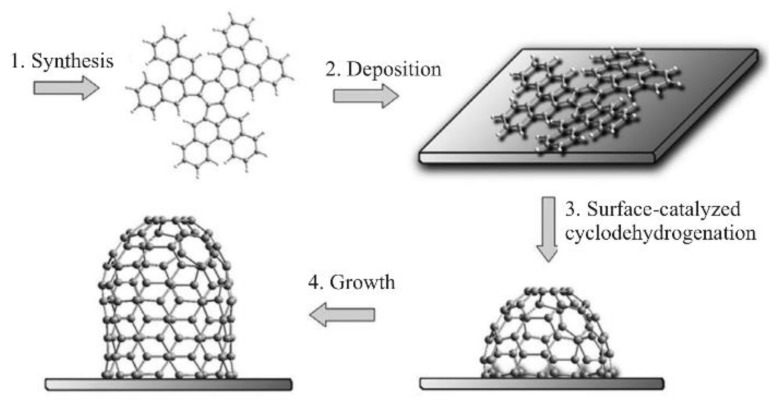
General scheme for the cyclodehydrogenation of the end-cap precursor and the subsequent growth of the CNT [69]. Copyright 2010, with permission from Elsevier.

**Figure 12 materials-15-05898-f012:**
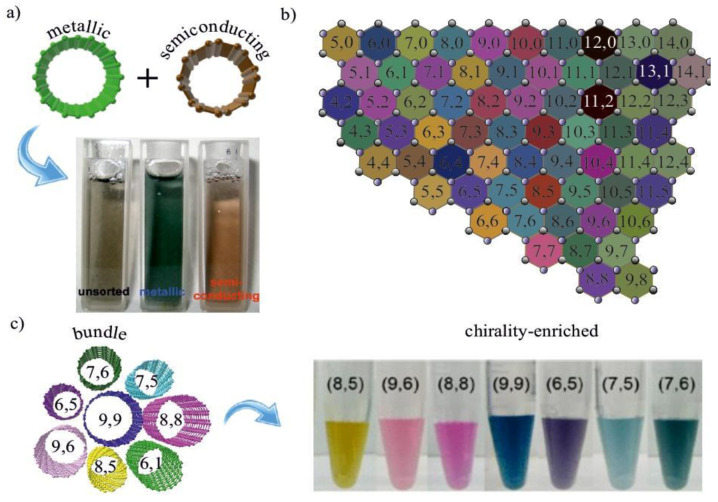
(**a**) The various types of carbon nanotubes are present in the unsorted samples and the corresponding photograph. (**b**) A map illustrates the colors of different types of carbon nanotubes with unique chiral indexes (n,m). (**c**) A mixture of different chirality SWCNTs in a bundle. Photograph of a few purified DNA-dispersed single-chirality SWCNTs [83]. Copyright 2016, American Chemical Society.

**Figure 13 materials-15-05898-f013:**
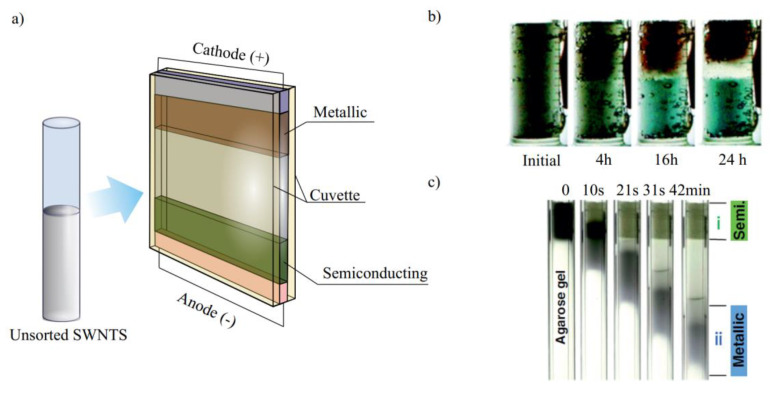
(**a**) Schematic of a vertical electrophoretic cell for a SWCNT separation. Metallic SWCNT (brown) having a high dielectric constant moves towards an electrode for the high applied electric field. Lower dielectric constant semiconducting SWCNT (green) is left in the solution. (**b**) Optical pictures taken before, 4, 16, and 24 h after the application of voltage on the SWCNT solution in a vertical cell (30 V). Nanotubes have a diameter of 1.0 nm [90]. Copyright 2011, American Chemical Society. (**c**) Separation of SWCNTs dispersed in gel by AGE. Optical images showing the progress of separation [86]. Copyright 2008, American Chemical Society.

**Figure 14 materials-15-05898-f014:**
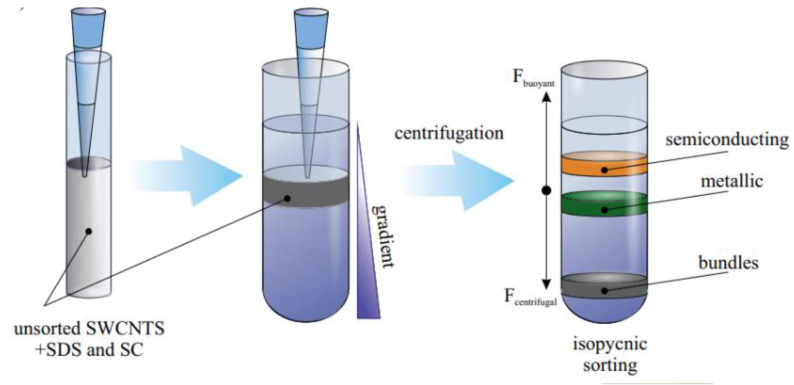
Schematics for the separation of SWCNTs using the density gradient ultracentrifugation technique. SWCNTs that have not been sorted are put into a centrifuge tube that has a gradient of mass density. During ultracentrifugation, due to a competition between the applied centrifugal force and the resultant opposing buoyant force, surfactant-wrapped SWCNTs migrate through the gradient until reaching the isopycnic point (equilibrium), where forces are balanced. The local gradient of the density gradient and the density of SWCNT are equivalent at this site.

**Figure 15 materials-15-05898-f015:**
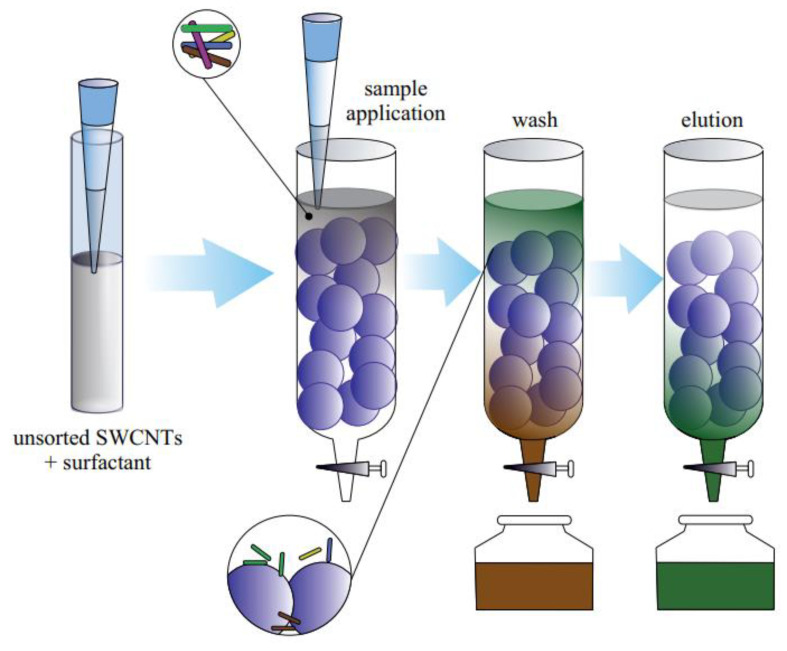
Gel chromatography, also called gel filtration, separates SWCNTs according to their electronic type. When unsorted SWCNTs are passed through a column packed with a matrix of porous beads, semiconducting SWCNTs flow through and around the beads in the direction of solvent flow, and metallic SWCNTs flow around the beads without interacting with the matrix material. Each of the fractions is collected after washing.

**Figure 16 materials-15-05898-f016:**
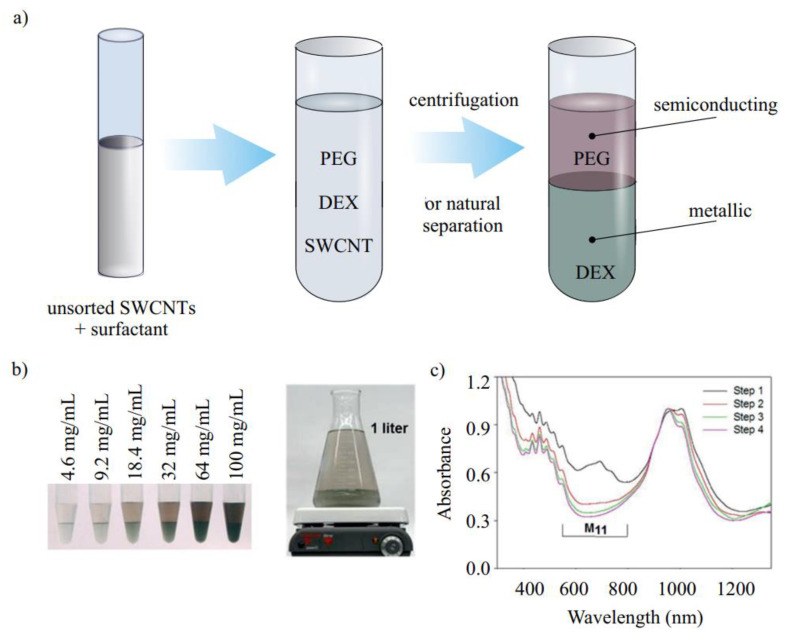
(**a**) Schematic image of two-phase extraction. The PEG, DEX, SWCNTs with cosurfactants are mixed. After the centrifugation or several hours of natural separation, the distinct layers at the top and bottom phases were observed. (**b**) The scalable and highly reproducible sorting of SWCNT by electronic type. The amount of CNT loading (from 4.6 to 100 mg/mL) is written. A 1 L volume partition sample. (**c**) Absorption spectra of semiconducting CNT fractions obtained from four steps of repeated partition [105]. Copyright 2013, American Chemical Society.

**Figure 17 materials-15-05898-f017:**
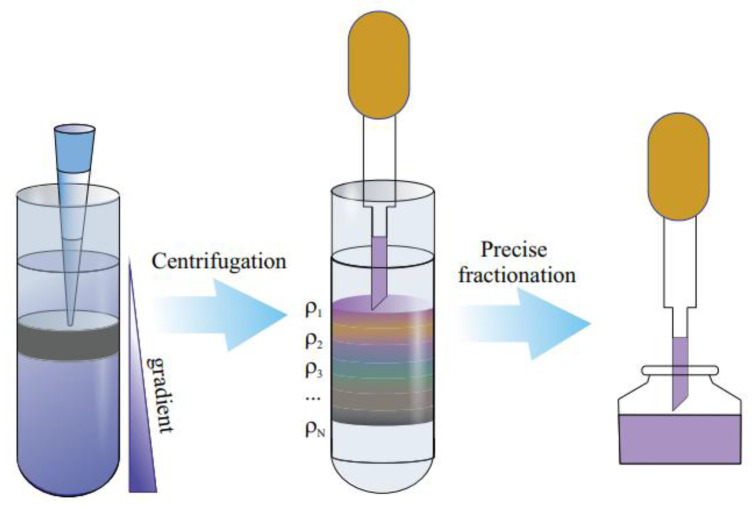
Schematics for the separation of SWCNTs using the density gradient ultracentrifugation technique. SWCNTs that have not been sorted are put into a centrifuge tube that has a mass density gradient at the isopycnic sites of the materials. Surfactant-wrapped SWCNTs move across a gradient to form separate colored layers during ultracentrifugation as a result of a conflict between the applied centrifugal force and the resulting opposing buoyant force. For the collection of chirality-enriched layers, a precise fractionation is necessary.

**Figure 18 materials-15-05898-f018:**
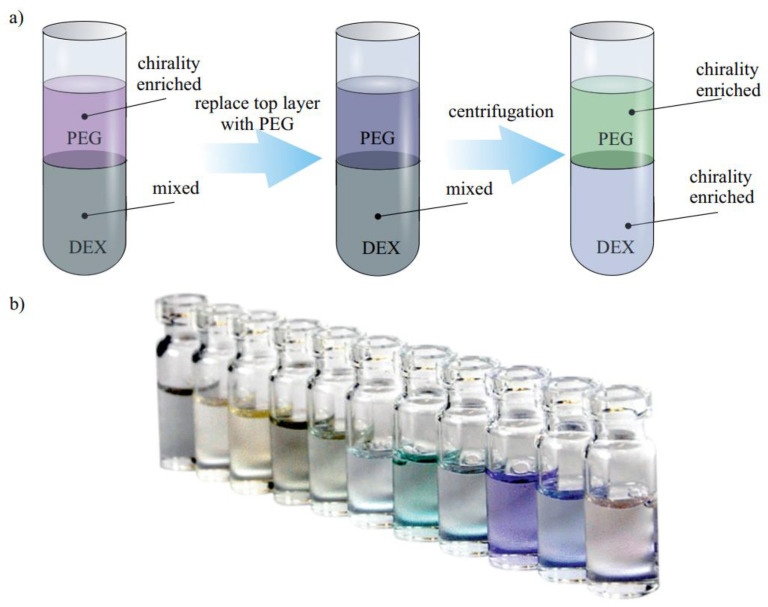
(**a**) Schematic illustration of two-phase separation for the multistep enrichment of single-chirality SWCNTs. (**b**) Optical photographs showing their distinct colors of 10 single-chirality SWCNTs achieved within three steps. When PEG and DEX solutions are mixed and two distinct aqueous phases form spontaneously, each has a different affinity towards SWCNTs. The phase enriched with single-chirality SWCNTs then is replaced by fresh PEG or DEX until the full separation is achieved [128]. Copyright 2019, American Chemical Society.

**Figure 19 materials-15-05898-f019:**
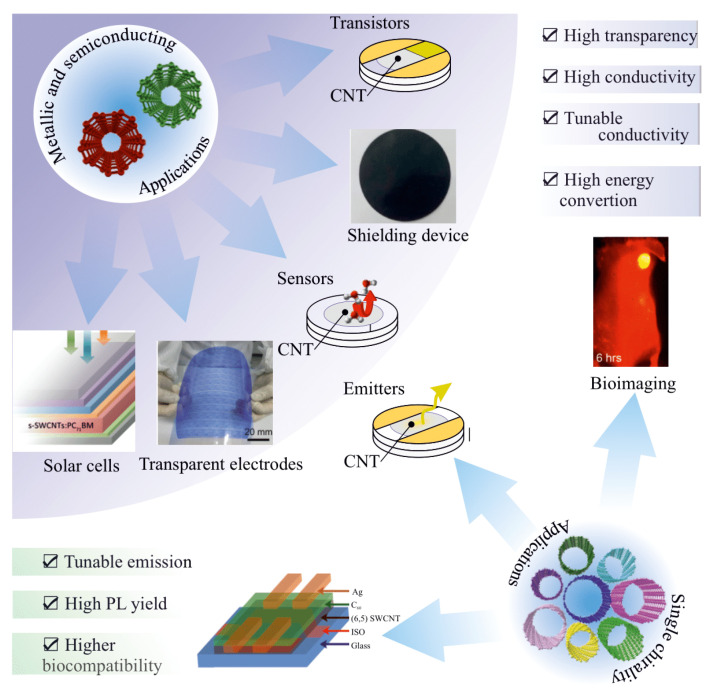
Overview of sorted SWCNT applications and benefits of using these materials. Sorted pure semiconducting and metallic SWCNTs can be used as a part of transistors, shielding devices, sensors, transparent electrodes, and solar cells. Meanwhile, chirality-enriched SWCNTs can be used for bioimaging, emitters, and solar cells.

**Figure 20 materials-15-05898-f020:**
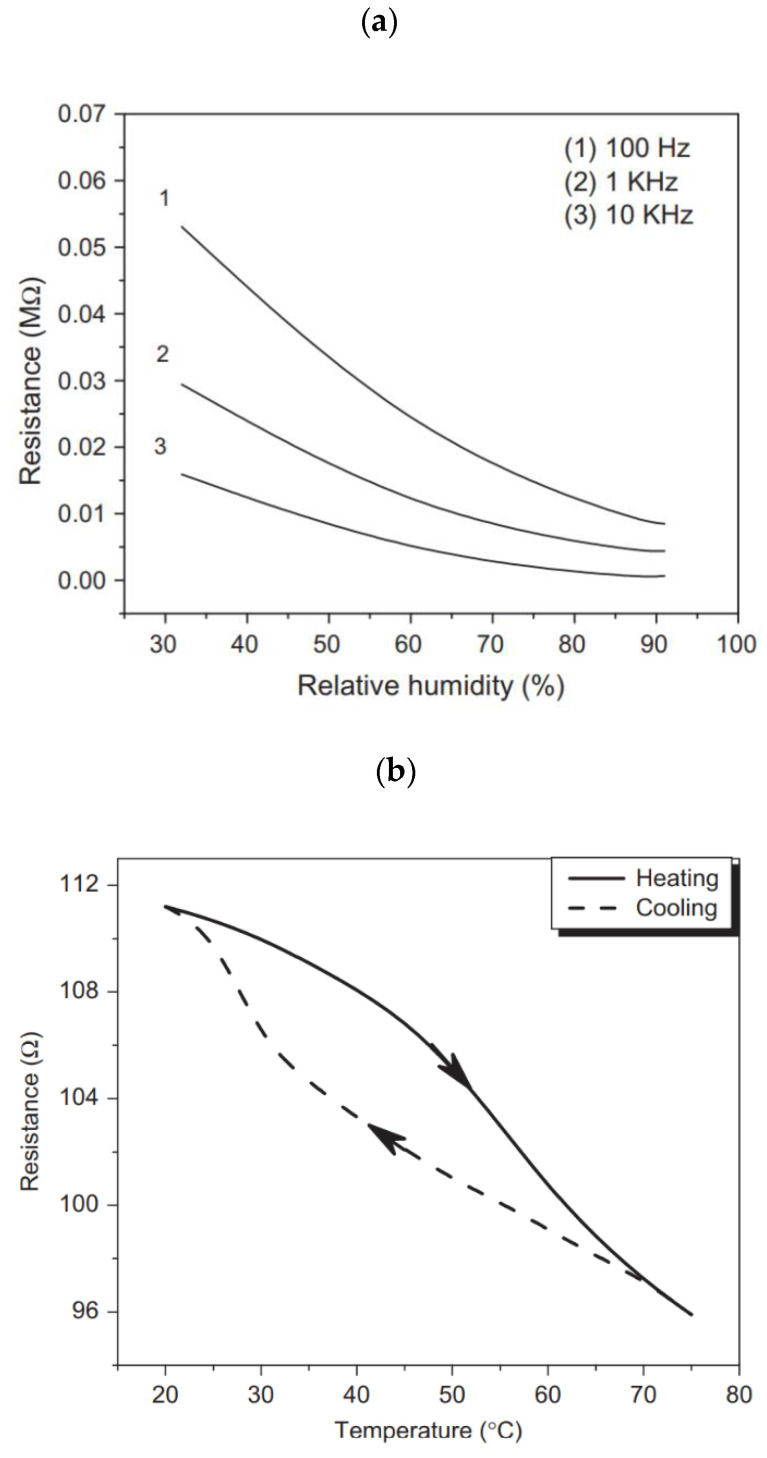
(**a**) Resistance-relative humidity relationships of the Ag/CNT/NiPc/PEPC/Au humidity sensor based on sorted SWCNTs. (**b**) Resistance–temperature relationships of one of the CNT-film-based sensors in heating–cooling processes [140]. Copyright 2011, with permission from Elsevier.

**Figure 21 materials-15-05898-f021:**
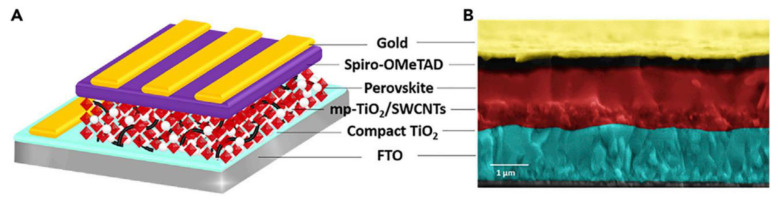
(**A**) Schematic diagram of the structure of PSCs, where CNTs (black lines) were incorporated into the TiO_2_ mesoporous scaffold layer. (**B**) Cross-sectional SEM image of the complete device [171]. Copyright 2019, with permission from Elsevier.

**Figure 22 materials-15-05898-f022:**
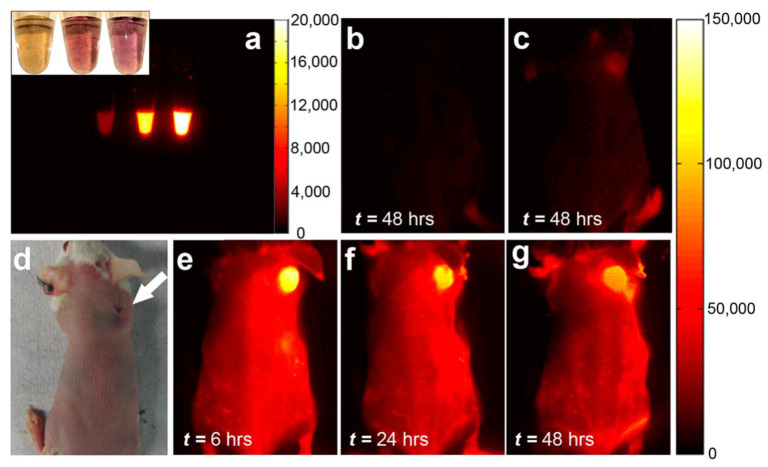
Time course fluorescence imaging and NIR (6,5) photoluminescence in the NIR-II. (**a**) An optical and NIR PL image of 50 L of C18-PMH-mPEG dispersed as synthesized HiPCO, first iteration (6,5) and second iteration (6,5), taken in the order of left to right and at a concentration of 22.2 g/mL. (**b**,**c**) NIR-II fluorescence pictures of a mouse injected with PBS and HiPCO and carrying a 4T1 tumor. (**d**) A mouse with a 4T1 tumor on its right shoulder is depicted optically. (**e**–**g**) From left to right, NIR-II fluorescent time series imaging demonstrates distinct SWCNT accumulation in the 4T1 tumor at 12, 24, and 48 h after injection. All animals received an injection of 0.254 mg/kg of SWCNTs, and after stimulation at 808 nm with a laser power density of 0.14 W/cm^2^, emission was measured from 0.9 to 1.4 m. Each NIR image of the 4T1 tumor-bearing Balb/c mice is scaled to the right [141]. Copyright 2013, American Chemical Society.

**Table 1 materials-15-05898-t001:** Main methods of separation of CNTs with targeted type of sorting and obtained performance.

	Synthesis	Electrophoresis	DGC	Chromatography	ATPE
Enrichment	Chirality	M, S	M, S, chirality	M, S, chirality	M, S, chirality
Scalability	Small scale	Small scale	Moderate scale	Large scale	Large scale
Purity	Low	High	High	Low	High
Cost	High	Low	High	Low	High
Speed	Slow	Fast	Slow	Slow	Fast
Simplicity	Complicated	Simple	Complicated	Simple	Simple

## Data Availability

Not applicable.

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
