# Peer review of "Synthesis, Sorting, and Applications of Single-Chirality Single-Walled Carbon Nanotubes"

_materials, 2022, doi:10.3390/ma15175898_

Round 1

Reviewer 1 Report

The proposed review by Kharlamova et al covers the main areas of interest regarding obtaining single-chirality SWCNTs. The research area remains one of the primary hurdles facing SWCNT research and a significant impediment to their use in next generation technologies and is a subject highly worthy of a review. My research focuses in part on the liquid phase processing of SWCNTs (n.b. not the growth or cloning) and I have hands on experience with several techniques, most notably the aqueous 2-phase separation, and while I am fully aware of the other main approaches (synthesis, cloning, bottom-up synthesis), I defer to my fellow reviewers in matters where our opinions differ. I also note that Kharmalova’s 2016 review in Prog Mater Sci is a personal favourite, and I fully looked forward to reading this manuscript.

In short, the work is of a good standard, covers the area reasonably thoroughly, and its publication would be warmly received by the community even in the current state, however, I believe there are several modifications which the authors should undertake to elevate the work.

Most notably, the ordering of the sections doesnt seem intuitive to me. The cloning section (which immediately follows the introduction) intrinsically requires pre-sorted SWCNTs, and the cloning then increases the quantity of chirally-pure nanotubes, not to make them in the first place. As such, it would make more sense to discuss sorting procedures, before later discussing how these sorted SWCNTs can be used, including as seeds for improved production.

Additionally, other non-pre-sorted-SWCNTs are discussed, although the term cloning appears erroneous in some instances. For the fullerenes, it is a stretch, but one could argue for the use of “cloning” as the fullerene can be seen as part of the resultant nanotube. However, growth from nanodiamonds is not cloning (ie you do not produce more nanodiamonds). Notably, the cloning terminology is not used in ref 38 or 39, where it is more directly compared to catalyst-growth technologies.

Much of the review focuses on simply listing techniques and highlights from the papers cited, which feels like a real missed opportunity. A lot more benefit could be gained from drawing out the comparisons between the techniques and outlining the benefits and weaknesses. I would recommend taking each broad technique and making radar charts plotting several factors (e.g. scalability, chiral precision, range of chiralities, cost, speed, ease, etc) qualitatively ranked 1-5 for the different processes (cloning/selective growth/ /electrophoresis/DGT/chromatography/ATPE) in their current state of technological development.

Some of the figures used are challenging to read due to small text size in-text (e.g. Fig 4b) and the authors should select alternative figures to illustrate their points. Regarding figures, the quality for Fig 18 is poor, with poor cropping leading to partial white outlines on some photo sides, white block backgrounds of schematics overlay on coloured backgrounds, and poor alignment throughout. The whole figure should be reworked to a more professional standard. I would also consider an additional less-schematic figure in addition to Fig 18 to show the many applications of chirally pure SWCNTs combining figures from literature.

Finally, the conclusion section is very weak and lacks a real outlook on the area. The authors should draw comparisons between the techniques, state where they think the most promise lies and the major challenges that will be faced, as well as how they see the technological landscape changing with freely available mono-chiral SWCNTs in the near, mid and long term.

The English used throughout is of good quality, although several minor issues are present throughout, although these are small enough to not confuse the reader and will presumably be ironed out in proofing. The references are all suitable and not major areas to my knowledge have been omitted.

Author Response

Dear reviewer

Thank you for your time and comments. 

We have made the following changes: 

1) We agree that the order of sections is not intuitive. We moved down the clonning section. 

2) Thank you for noticing it. This is true. We removed the word "clonning" from the title of this section, leaving "2.2. The epitaxial growth method of SWCNT using nanocarbon seeds". We then went through this section and removed the word "clonning" where it was necessary.

3)Thank you for this comment. We have included a table which compares different parameters as you suggested. 

4) We have updated figures and added a table which describes the possible applications of sorted SWCNT as well as the comparison.

5) We have improved the quality and font size of some figures.

6) Finally, we added a new section: Remaining Challenges and Future outlook

Reviewer 2 Report

The authors report a comprehensive review of the recent advanced on single-chirality single- walled carbon nanotubes ranging from synthesis, sorting, and purification methods as well as wide range of applications in electronics, photonics and bioimaging. In general, it’s a well-written and organized survey that contains relevant information about the subject. Below, I point out some amendments and suggestions.

1)  The resolution of the Figures 1 and Figure 2 is poor and must be improved.

2) All the figure’s numeration and citation in the main text must be reviewed and corrected. For instance, Figures 11 and 13 are not mentioned in the main text and Figure 14a is wrongly cited.

3) Line 1066: The correct is “In order to” not “In order for”.

4) Line 1041. Section 4.4. Transparent electrodes. I strongly recommend the authors include a review considering the combination of SWCNT and silver nanowires for transparent electrodes. There are relevant publications and industrial initiatives demonstrating the success of this hybrid SWCNT/AgNW approach. Herein, a couple examples to get into the subject: https://doi.org/10.1016/j.compositesb.2018.06.004;

https://www.chasmtek.com/chasm-products.

5) Line 1147. Replace “99,9 %” by “99.9%”. 

Overall, the paper has quality enough to appear in Materials since all the corrections mentioned above be done.

Author Response

Dear Reviewer

Thank you for your comments. 

1) We improved the quality of figures. 

2) Thank you for spotting this. We checked and fixed all numerations of figures.

3) Thank you for noticing this. 

4) We have added a paragraph about AgNW-CNT composite.

5) Thank you for spotting this.

Reviewer 3 Report

This manuscript is a nice comprehensive review paper dedicated to the chirality selective synthesis, sorting of SWCNTs and applications of chirality pure 16 SWCNTs. Authors carefully discussed: 1) synthesis methods of semiconducting, metallic and single chirality SWCNTs; 2) separation of SWCNTs by conductivity type and chiral angle; 3) application of conductivity-type and chirality separated SWCNTs. The manuscript is written clearly and I believe it fits the scope of Materials. However, I have a few questions/comments to authors for further clarification and make some remarks regarding the manuscript and data presentation for a wider audience.

1.     I suggest the authors to add an overview diagram of this review as Figure 1. It’s like the current Fig. 18, but it should include synthesis, separation, and application of SWCNTs. This figure will give readers a clear overview of what will be discussed in this review.

2.     Figure panel layout should be consistent. In the current manuscript, authors randomly use “(a)”, “a”, “a)” as panel labels in different figures, which should be avoided.

3.     For the synthesis section, Fig. 5, 6, 7, 8, 9 all showed the photoluminescence results of the SWCNTs synthesized with various methods. Authors should consider to add the schematic diagrams of the different synthetic method or their design of the catalyst, not just show the PL results. The combination of synthetic design and growth results in the figures will give readers a better understanding of the synthetic methods.

Author Response

Dear Reviewer

Thank you for your comments. 

1) Thank you for this suggestion. We added the required figure.

2) Thank you for noticing it. We used (a) in all Figures.

3) We have a similar request from another reviewer and added a diagram at the beginning of the review.

Kind regards,

Maria